# Source apportionment of atmospheric water over East Asia – a source tracer study in CAM5.1

Chen Pan[1], Bin Zhu[1], Jinhui Gao[1], Hanqing Kang[1]

[1]Key Laboratory of Meteorological Disaster, Ministry of Education (KLME), Joint International Research Laboratory of Climate and Environment Change (ILCEC), Collaborative Innovation Center on Forecast and Evaluation of Meteorological Disasters, Key Laboratory for Aerosol-Cloud-Precipitation of China Meteorological Administration, Nanjing University of Information Science & Technology, Nanjing, 210044, China

*Correspondence to*: Bin Zhu (binzhu@nuist.edu.cn)

**Abstract**
The atmospheric water tracer (AWT) method is implemented in the Community Atmosphere Model version 5.1 (CAM5.1)
to quantitatively identify the contributions of various source regions to precipitation and water vapour over East Asia.
Compared to other source apportionment methods, the AWT method was developed based on detailed physical
parameterizations, and can therefore trace the behaviour of atmospheric water substances directly and exactly. According to
the simulation, the north Indian Ocean (NIO) is the dominant oceanic moisture source region for precipitation over the
Yangtze River Valley (YRV) and South China (SCN) in summer, while the Northwest Pacific (NWP) dominates during
other seasons. Evaporation over the South China Sea (SCS) is responsible for only 2.7–3.7% of summer precipitation over
the YRV and SCN. In addition, the Indo-China Peninsula is an important terrestrial moisture source region (annual
contribution of ~10%). The overall relative contribution of each source region to the water vapour amount is similar to the
corresponding contribution to precipitation over the YRV and SCN. A case study for the SCS shows that only a small part
(≤5.5%) of water vapour originates from local evaporation, while much more water vapour is supplied by the NWP and NIO.
In addition, because evaporation from the SCS represents only a small contribution to the water vapour over the YRV and
SCN in summer, the SCS mainly acts as a water vapour transport pathway where moisture from the NIO and NWP meet.
**Keywords**
Atmospheric water tracer method; Community Atmosphere Model; source apportionment; precipitation and water vapour

## 1 Introduction

Water vapour is one of the most important components of the atmosphere, affecting global climate and weather patterns (Held and Soden, 2000). Among current studies of the hydrological cycle, the identification of moisture sources to the atmosphere is an important topic, because a better understanding of these sources will benefit long-term forecasting, disaster prevention, and allocation of water resources (Bosilovich and Schubert, 2002).

Source apportionment methods have been developed to identify atmospheric moisture source regions. These methods generally can be divided into three types, namely analytical models, isotopes, and numerical (Lagrangian and Eulerian) atmospheric water tracers (AWTs) (Gimeno et al., 2012). In addition, sensitivity experiments in numerical simulations such as shutting down water vapour flux at the lateral boundaries or surface evaporation (Chow et al., 2008) are an approach to study the contributions of moisture from diverse regions. Analytical models, widely used in earlier studies (Brubaker et al., 1993; Burde and Zangvil, 2001; Eltahir and Bras, 1996; Savenije, 1995; Trenberth, 1999), are generally based on various simplifying assumptions such as a well-mixed atmosphere. The stable isotopes of water, HDO and $H_2^{18}O$, can be used to investigate the water cycle. However, water isotope data reflect a series of processes that occur simultaneously, which makes it difficult to interpret isotope results for the water cycle (Numaguti, 1999; Sodemann and Zubler, 2010). The Lagrangian method has become a popular way to analyse the transport of moisture and moisture sources of precipitation (Dirmeyer and Brubaker, 1999; Gustafsson et al., 2010; Sodemann et al., 2008; Stohl and James, 2004; Stohl et al., 2008). However, Gimeno et al. (2012) pointed out that the treatments of water vapour transport and changes of atmospheric water vapour in the Lagrangian method are not based on detailed physical equations. Sodemann and Zubler (2010) pointed out that a strong bias exists in Lagrangian precipitation estimates, because all cloud processes are neglected. Sensitivity experiments generally contain nonlinearities, which may lead to changes in the dynamic and thermodynamic structures of meteorological fields, suggesting that their results cannot be used to directly diagnose moisture sources. In contrast, the Eulerian AWT method has been developed based on detailed physical parameterizations in atmospheric models, enabling a direct and exact tracking of the behaviour of atmospheric water substances (Numaguti, 1999; Bosilovich, 2002).


The Eulerian AWT method was firstly developed by Joussaume et al. (1986) and Koster et al. (1986) for global circulation
models (GCMs). Later, this AWT method was applied to diagnose regional water sources in GCMs. For example, Numaguti
(1999) identified the moisture sources of Eurasian precipitation, and Bosilovich and Schubert (2002) diagnosed the moisture
sources of precipitation over North America and India. Bosilovich et al. (2003) studied water sources of the large-scale
North American monsoon, Bosilovich (2002) investigated the vertical distribution of water vapour tracers over North
America, and Sodemann et al. (2009) used this method to study sources of water vapour leading to a flood event in Central
Europe using a mesoscale model. Finally, Knoche and Kunstmann (2013) incorporated the AWT method into a fifth-
generation mesoscale model to study the transport of atmospheric moisture in West Africa.

In summer, the Asian summer monsoon (ASM) brings large amounts of water vapour to the East Asian (EA) continent,
leading to a wet season and abundant precipitation. Simmonds et al. (1999) pointed out that the dominant moisture transport
pathways during summer can be divided into three branches, namely (i) southwesterly flow associated with the Indian
summer monsoon, (ii) southerly or southeasterly flow associated with the southeastern Asian monsoon, and (iii) the mid-
latitude Westerlies. Correspondingly, these pathways transport moisture from (i) the Bay of Bengal (BOB) and the Arabian
Sea (AS), (ii) the South China Sea (SCS) and the Northwest Pacific (NWP), and (iii) the mid-latitude regions. Simmonds et
al. (1999) and Xu et al. (2008) pointed out that the BOB to SCS are the main source regions for rainfall over southeast China.
Using the Lagrangian Flexible Particle (FLEXPART) dispersion model (Stohl and James, 2004), Drumond et al. (2011)
discovered that the inland regions of China receive moisture mostly from western Asia, while the East China Sea (ECS) and
SCS are the main source regions for rainfall in China's eastern and southeastern coastal areas and the AS and BOB are the
main source regions for southern and central China from April to September. With the FLEXPART model, Baker et al.
(2015) demonstrated that the Indian Ocean is the primary source of moisture for East Asian summer monsoon (EASM)
rainfall. Using the same model, Chen et al. (2013) suggested that the ECS, the SCS, the Indian peninsula and BOB, and the
AS were the four major moisture source regions for summer water vapour over the Yangtze River Valley (YRV) during
2004–2009. Chow et al. (2008) suggested that water vapour supplied by the Indian summer monsoon contributed about 50%
to early summer precipitation over China in 1998, and inferred that the SCS may act as a pathway for water vapour transport
affected by the Indian and Southeast Asian summer monsoon. However, recently Wei et al. (2012), using a Lagrangian
model, showed that the major moisture transport pathways to the YRV are over land and not over the ocean. Therefore, the
dominant source regions of moisture for summer rainfall over EA are still uncertain.

Baker et al. (2015) pointed out that the water vapour transport mechanisms for precipitation over China during the ASM are
still unquantified. Previous studies have pointed out that analytical models need simplifying assumptions, isotope data
reflects more than just the water cycle, the Lagrangian methods lack cloud processes, and sensitivity experiments contain
nonlinearities, limiting diagnostic studies of moisture sources. On the other hand, the Eulerian AWT method does not have
these shortcomings and is an accurate way to quantitatively determine water sources (Bosilovich, 2002). Therefore, in this
study, we aim at incorporating an Eulerian AWT approach into an advanced global atmosphere model – the Community
Atmosphere Model version 5.1 (CAM5.1) (Neale et al., 2012). Using this method, we address the following questions: (1)
What moisture source regions are most important for precipitation and water vapour amount over EA, including the YRV
and South China (SCN)? (2) What is the role of the SCS for precipitation and water vapour amount over EA during the
EASM: a dominant source region or just a pathway for water vapour transport from other source regions?

In this study, detailed descriptions of physical parameterization schemes and means of implementing the AWT mechanisms
in CAM5.1 are given in Sect. 2. Simulation results, including evaluation and discussion, are presented in Sect. 3. Finally,
summary and concluding remarks are presented in Sect. 4.

**2 Model and methods**
The CAM5.1, released by the U.S. National Center for Atmospheric Research, is the atmospheric component of the
Community Earth System Model (CESM) (Neale et al., 2012). Compared to CAM4, CAM5.1 contains a range of
improvements in the representation of physical processes such as moist turbulence, shallow convection, stratiform
microphysics, cloud macrophysics schemes, and others (Neale et al., 2012). The horizontal resolution used in this study is
1.9 °in latitude and 2.5 °in longitude. The vertical range is from the surface to approximately 4 hPa ($\approx$ 40 km).

In this study, the chemistry mechanism of CAM5.1 is taken from MOZART-4 (Emmons et al., 2010), in which water vapour
is invariant, which means that it is unnecessary to consider changes in water vapour during chemical processes. The basic
simulations setup, including emissions and upper and lower boundary conditions, is identical to that of the specified
dynamics simulations of CAM5 in Lamarque et al. (2012). In this study, the wet removal scheme in Horowitz et al. (2003) is
adopted. The temporal evolution of the mass mixing ratios (MMRs) of different water substances (water vapour, cloud
droplets, and ice) is determined by deep convection, shallow convection, cloud macrophysics, cloud microphysics, advection,
and vertical diffusion. To diagnose the dominant moisture source regions of atmospheric water over EA, the global surface is
divided into 25 source regions as shown in Fig. 1. Most regions are defined based on the locations of continents and oceans.
Due to the focus on moisture sources over EA in this study, EA and its adjacent regions are further divided to provide more
detail. Within source region $k$, the surface flux of the tagged water vapour tracer $E^k$ is equal to the surface evaporation flux
of water vapour $E$; otherwise $E^k = 0$. As in the treatment described in Knoche and Kunstmann (2013) and Bosilovich and
Schubert (2002), water is "tagged" when it evaporates at its source region and is no longer tagged when it precipitates from
the atmosphere to the Earth's surface via atmospheric processes. When previously tagged precipitation reevaporates from the
surface, it is regarded as newly tagged water (Knoche and Kunstmann, 2013), which then belongs to the region from where it
reevaporates.

The MMRs of water vapour, cloud droplets, and ice at a particular level are defined as $q_v$, $q_l$, and $q_i$, respectively. The
corresponding MMRs of tagged water substances from source region $k$ are $q_{v,tg}^k$, $q_{l,tg}^k$, and $q_{i,tg}^k$. We assume that all the
tagged water substances from the source regions have the identical physical properties and are well-mixed. All these tagged
water substances are passive, which means that they are entirely separate from the original water substances in CAM5.1 and
have no impact on dynamical and thermal fields. Numaguti (1999) suggested that the lifetime of atmospheric water vapour is
about 10 days. In this study, the simulation is started in 01 January 1997, and the initial MMRs of tagged substances are set
to zero. To attain stable initial concentrations of tagged water substances, the simulation experiment takes a year to spin up.
We then investigate the ten-year averaged results for 1998 to 2007. In the following, we describe the treatment of tagged
AWTs in CAM5.1's physical parameterizations.

## 128    2.1 Deep convection

In CAM5.1, deep convection is parameterized using the approach described in Zhang and McFarlane (1995), but with
modifications following Richter and Rasch (2008) and Raymond and Blyth (1986, 1992). For the temporal evolution of $q_{v,tg}^{k}$,
it is calculated in the same way as that of $q_v$, but the relevant variables of tagged water vapour are substituted for the
corresponding variables of original water vapour, expressed as:
$$\left(\frac{\partial q_{v,tg}^{k}}{\partial t}\right)_{\text{dp}} = \epsilon_{tg}^{k} - c_{tg}^{k} - \frac{1}{\rho}\frac{\partial}{\partial z}\left(M_{u,\text{dp}}q_{v,u,tg}^{k} + M_{d,\text{dp}}q_{v,d,tg}^{k} - M_{c,\text{dp}}q_{v,tg}^{k}\right) \tag{1}$$
where $M_{c,\text{dp}}$ is the net vertical mass flux, $M_{u,\text{dp}}$ is the upward mass flux, and $M_{d,\text{dp}}$ is the downward mass flux in the deep
convection. $\epsilon_{tg}^{k}$ and $c_{tg}^{k}$ are the large-scale mean evaporation and condensation rates of tagged water vapour, respectively.
Here, $q_{v,u,tg}^{k}$ and $q_{v,d,tg}^{k}$ are the MMR of tagged water vapour in the updraft and that in the downdraft, respectively. The ratio
between the MMR of tagged water vapour and the corresponding sum is used to calculate the condensation rate $c_{tg}^{k}$:
$$c_{tg}^{k} = \left(\frac{q_{v,tg}^{k}}{\sum_{k=1}^{n} q_{v,tg}^{k}}\right)c \tag{2}$$
where $c$ is the condensation of original water vapour. In this study, n=25, which is the total number of defined source regions
(Fig. 1). In this scheme, the tagged cloud water in the updraft, the detrainment of tagged cloud water, rain production rate,
and the evaporation rate of tagged rain in the downdraft are calculated in the same manner as that for the corresponding
quantities for original water. However, the relevant variables of tagged water vapour are substituted for the corresponding
variables of original water vapour. Detailed formulas for relevant quantities for original water in the updraft and downdraft
are presented in Sect. 3 of Zhang and McFarlane (1995). The evaporation of convection precipitation is also considered in
this parameterization. The evaporation rate $\left(\frac{\partial q_v^k}{\partial t}\right)_{\text{dp\_evap}}$ at level $m$ is associated with the deep convection precipitation flux
$(Q_m)_{\text{dp}}$ at the top interface of this level (Sundqvist, 1998), expressed as
$$\left(\frac{\partial q_v^k}{\partial t}\right)_{\text{dp\_evap}} = k_e(1 - \text{RH}_m)\sqrt{(Q_m)_{\text{dp}}} \tag{3}$$
where $\text{RH}_m$ is the relative humidity at level $m$ and the coefficient $k_e = 2 \times 10^{-6}$ (kg m$^{-2}$ s$^{-1}$)$^{-1/2}$ s$^{-1}$. The individual
evaporation rate of tagged convection precipitation from source region $k$ is calculated as:
$$\left(\frac{\partial q_{v,tg}^k}{\partial t}\right)_{\text{dp\_evap}} = \begin{cases} k_e(1 - \text{RH}_m)\dfrac{\left(Q_{m,tg}^k\right)_{\text{dp}}}{\sqrt{\sum_{k=1}^n \left(Q_{m,tg}^k\right)_{\text{dp}}}}, & \text{if } \sum_{k=1}^n \left(Q_{m,tg}^k\right)_{\text{dp}} \neq 0, \\ 0, & \text{if } \sum_{k=1}^n \left(Q_{m,tg}^k\right)_{\text{dp}} = 0 \end{cases} \tag{4}$$
In general, the evaporation rate of convection precipitation is very small compared to the tendency of water vapour in the
deep convection (Neale et al., 2012). For the temporal evolution of $q_{l,tg}^k$ and $q_{i,tg}^k$ in the deep convection parameterization,
both are treated in the same subroutine as $q_l$ and $q_i$.

**2.2 Shallow convection**
The shallow convection scheme in CAM5.1 is taken from Park and Bretherton (2009). Similar to the MMR of the total water
$q_t$, the MMR of the tagged total water $q_{t,tg}^k$ is also assumed to be a conserved quantity in non-precipitating moist adiabatic
processes. In this scheme, the diagnostic equations for the shallow convective mass flux $M_{u,\text{sh}}$ and the MMR of the updraft
total water $q_{t,u}$ (Bretherton et al., 2004) are expressed as:
$$\frac{\partial M_{u,\text{sh}}}{\partial z} = E_{tr} - D_{tr} \tag{5}$$
and
$$\frac{\partial}{\partial z}\left(q_{t,u} M_{u,\text{sh}}\right) = E_{tr}\bar{q}_t - D_{tr} q_{t,u} + \left(\frac{\partial q_t}{\partial z}\right) M_{u,\text{sh}} \tag{6}$$
where $E_{tr}$ is the entrainment rate, $D_{tr}$ is the detrainment rate, and $\bar{q}_t$ is the MMR of the mean environmental total water. The
fractional entrainment and detrainment rates are denoted as $\varepsilon$ and $\delta$, then
$E_{tr} = \varepsilon M_{u,\text{sh}}, D_{tr} = \delta M_{u,\text{sh}}$ (7)
Finally, attaining the updraft dilution equations:
$\frac{\partial M_{u,\text{sh}}}{\partial z} = M_{u,\text{sh}}(\varepsilon - \delta)$ (8)
$\frac{\partial q_{t,u}}{\partial z} = \varepsilon(\bar{q}_t - q_{t,u}) + \frac{\partial q_t}{\partial z}$ (9)
Similarly, the updraft dilution equation for the tagged total water is expressed as:
$\frac{\partial q_{t,u,tg}^k}{\partial z} = \varepsilon(\bar{q}_{t,tg}^k - q_{t,u,tg}^k) + \frac{\partial q_{t,tg}^k}{\partial z}$ (10)
Equation (A5) of Bretherton et al. (2004) is used to calculate $q_{t,u}$, as well as $q_{t,u,tg}^k$, in the shallow convection. In this scheme,
because the detrainment of cloud water and ice ($D(q_l)$ and $D(q_i)$) is assumed to be proportional to the total water
detrainment and the detrained air is assumed to be a representative of cumulus updraft (Park and Bretherton, 2009), we use
the ratio of tagged total water in the updraft $q_{t,u,tg}^k$ and the corresponding sum to distribute the detrainment of tagged cloud
water and ice ($D(q_{l,tg}^k)$ and $D(q_{i,tg}^k)$):
$D(q_{l,tg}^k) = \left(\frac{q_{t,u,tg}^k}{\sum_{k=1}^{n} q_{t,u,tg}^k}\right) \times D(q_l), \ D(q_{i,tg}^k) = (\frac{q_{t,u,tg}^k}{\sum_{k=1}^{n} q_{t,u,tg}^k}) \times D(q_i)$ (11)
This ratio is also applied to the calculations of in-cumulus tagged condensates and the production rates of tagged rain/snow
by cumulus expulsion of condensates to the environment. Tagged condensate tendencies for compensating subsidence or
upwelling, the tagged condensate tendencies due to detrained cloud water and ice without precipitation contribution, and the
updraft/penetrative entrainment mass flux of tagged total water are calculated using the same equations for the original
water-related quantities in this scheme. Similar to the calculation of the tendency of water vapour, the tendency of tagged
water vapour is computed as the difference between the tendency of tagged total water and the tendencies of tagged
condensates in non-precipitating processes within the shallow convection scheme. The shallow convection scheme relates
precipitation evaporation rate $\left(\frac{\partial q_v}{\partial t}\right)_{\text{sh\_evap}}$ to shallow convection precipitation flux $Q_{\text{sh}}$, similar to the deep convection
scheme of CAM5.1. Therefore, we use an assumed expression similar to Eq. (4) to calculate the tagged precipitation
evaporation rate at a level $m$:
$$\left(\frac{\partial q_{v,tg}^k}{\partial t}\right)_{\text{sh\_evap}} = \begin{cases} k_e(1 - \text{RH}_m)\dfrac{\left(Q_{m,tg}^k\right)_{\text{sh}}}{\sqrt{\sum_{k=1}^{n}\left(Q_{m,tg}^k\right)_{\text{sh}}}}, & \text{if } \sum_{k=1}^{n}\left(Q_{m,tg}^k\right)_{\text{sh}} \neq 0 \\[4mm] 0, & \text{if } \sum_{k=1}^{n}\left(Q_{m,tg}^k\right)_{\text{sh}} = 0 \end{cases} \quad (12)$$
where $\left(Q_{m,tg}^k\right)_{\text{sh}}$ is the tagged precipitation flux at the top interface of level $m$.

**2.3 Cloud Macrophysics**
Park et al. (2014) provided a detailed description of CAM5.1's cloud macrophysics, in which cloud fractions, horizontal and
vertical overlapping structures of clouds, and net condensation rates of water vapour into cloud droplets and ice are
computed. Because the tendencies of water substances caused by cumulus convection have been calculated in deep and
shallow convection schemes, we focus on the treatment of the tagged stratus fraction and net condensation rates of tagged
water vapour in stratus clouds in this section.

The separate liquid stratus fraction $a_{l,\text{st}}$ is a unique function of grid-mean relative humidity (RH) over water, $\bar{u}_l \equiv \bar{q}_v/\bar{q}_{s,w}$,
where $\bar{q}_v$ is the grid-mean water vapour specific humidity and $\bar{q}_{s,w}$ is the grid-mean saturation specific humidity over water,
which is shown in Eq. (3) of Park et al. (2014). Then the single-phase (no separate liquid and ice phases) liquid stratus
fraction is
$A_{l,\text{st}} = (1 - A_{\text{cu}})a_{l,\text{st}}$ (13)
Here $A_{\text{cu}}$ is the total cumulus fraction.
We allocate the tagged liquid stratus fraction $A_{l,\text{st},tg}^k$, which depends on the ratio of grid-mean tagged water vapour specific
humidity $\bar{q}_{v,tg}^k$ and the corresponding sum, expressed as:
$A_{l,\text{st},tg}^k = \left(\dfrac{\bar{q}_{v,tg}^k}{\sum_{k=1}^{n}\bar{q}_{v,tg}^k}\right)A_{l,\text{st}}$ (14)
The tagged grid-mean liquid stratus condensate $\bar{q}_{l,\text{a},tg}^k$ is calculated in the same way as the grid-mean liquid stratus
condensate $\bar{q}_{l,\text{a}}$, but $A_{l,\text{st},tg}^k$ is substituted for $A_{l,\text{st}}$:
$\bar{q}_{l,a,tg}^{k} = A_{l,\text{st},tg}^{k} \times q_{l,\text{st}}$ (15)
Here, $q_{l,\text{st}}$ is the in-stratus liquid water content (LWC). Similar to $a_{l,\text{st}}$, the ice stratus fraction $a_{i,\text{st}}$ is a function of the grid-
mean total ice RH over ice, $\bar{v}_i \equiv (\bar{q}_v + \bar{q}_i)/\bar{q}_{s,i}$, where $\bar{q}_i$ is the grid-mean ice specific humidity and $\bar{q}_{s,i}$ is the grid-mean
saturation specific humidity over ice, as shown in Eq. (4) of Park et al. (2014). Similar to $A_{l,\text{st}}$, the single-phase ice stratus
fraction is calculated as
$A_{i,\text{st}} = (1 - A_{\text{cu}})a_{i,\text{st}}$ (16)
As in the treatment of $A_{l,\text{st},tg}^{k}$, the tagged ice stratus fraction $A_{i,\text{st},tg}^{k}$ is computed based on the ratio of grid-mean total tagged
ice specific humidity $(\bar{q}_{v,tg}^{k} + \bar{q}_{i,tg}^{k})$ and the corresponding sum:
$A_{i,\text{st},tg}^{k} = \left[ \dfrac{(\bar{q}_{v,tg}^{k} + \bar{q}_{i,tg}^{k})}{\sum_{k=1}^{n}(\bar{q}_{v,tg}^{k} + \bar{q}_{i,tg}^{k})} \right] A_{i,\text{st}}$ (17)
The tagged grid-mean ice stratus condensate $\bar{q}_{i,a,tg}^{k}$ is calculated in the same way as the grid-mean ice stratus condensate $\bar{q}_{i,a}$:
$\bar{q}_{i,a,tg}^{k} = A_{i,\text{st},tg}^{k} \times q_{i,\text{st}}$ (18)
Here, $q_{i,\text{st}}$ is the in-stratus ice water content (IWC). Using the same formula as for the calculation of the grid-mean ambient
water vapour specific humidity, the tagged grid-mean ambient water vapour specific humidity $\bar{q}_{v,a,tg}^{k}$ is computed as follows:
$\bar{q}_{v,a,tg}^{k} = \bar{q}_{v,tg}^{k} + \bar{q}_{l,tg}^{k} + \bar{q}_{i,tg}^{k} - \bar{q}_{l,a,tg}^{k} - \bar{q}_{i,a,tg}^{k}$ (19)

In CAM5.1, Park et al. (2014) defined the grid-mean net condensation rate of water vapour into liquid stratus condensate $\bar{Q}_l$
as the time change of $\bar{q}_{l,\text{a}}$ minus the external forcing (all processes except stratus macrophysics, including stratus
microphysics, moisture turbulence, advection, and convection) of cloud droplets $\bar{F}_l$:
$\bar{Q}_l = \dot{\bar{q}}_{l,a} - \bar{F}_l = A_{l,\text{st}}\dot{q}_{l,\text{st}} + \alpha q_{l,\text{st}}\dot{A}_{l,\text{st}} - \bar{F}_l$ (20)
where $\dot{\bar{q}}_{l,a}$, $\dot{q}_{l,\text{st}}$, and $\dot{A}_{l,\text{st}}$ are the time tendency of $\bar{q}_{l,\text{a}}$, $q_{l,\text{st}}$, and $A_{l,\text{st}}$ during $\Delta t = 1800$ s, respectively. In CAM5.1, $\alpha = 0.1$
is the ratio of newly formed or dissipated stratus to the preexisting $q_{l,\text{st}}$. Similarly, the tagged grid-mean net condensation
rate $\bar{Q}_{l,tg}^{k}$ is calculated as:
$\bar{Q}_{l,tg}^{k} = \dot{\bar{q}}_{l,a,tg}^{k} - \bar{F}_{l,tg}^{k} = A_{l,\text{st},tg}^{k}\dot{q}_{l,\text{st}} + \alpha q_{l,\text{st}}(R\dot{A}_{l,\text{st}} + A_{l,\text{st}}\dot{R}) - \bar{F}_{l,tg}^{k}, \text{and } R = \dfrac{\bar{q}_{v,tg}^{k}}{\sum_{k=1}^{n}\bar{q}_{v,tg}^{k}}$ (21)
Here, $\dot{R}$ is the tendency of $R$ during $\Delta t$, and $\bar{F}_{l,tg}^{k}$ is the changes of tagged cloud droplets in processes such as microphysics,
moisture turbulence, advection, and deep and shallow convections.

## 2.4 Cloud Microphysics

The CAM5.1 model uses the double-moment cloud microphysical scheme described in Morrison and Gettelman (2008) and
a modified treatment of ice supersaturation and ice nucleation from Gettelman et al. (2010). In addition, CAM5.1's stratus
microphysics is formulated using a single-phase stratus fraction $A_{\text{st}}$, which is assumed as the maximum overlap between
$A_{l,\text{st}}$ and $A_{i,\text{st}}$ (Park et al., 2014). In this study, the same assumption is applied to each tagged single-phase stratus fraction
$A_{\text{st},tg}^{k}$. The microphysical processes in CAM5.1 include condensation/deposition, evaporation/sublimation, autoconversion of
cloud droplets and ice to form rain and snow, accretion of cloud droplets and ice by rain or by snow, heterogeneous freezing,
homogeneous freezing, melting, sedimentation, activation of cloud droplets, and primary ice nucleation. Detailed
formulations for these microphysical processes are described in Morrison and Gettelman (2008).

### 2.4.1 Condensation/deposition and evaporation/sublimation of cloud water and ice

In CAM5.1, the net grid-mean evaporation/condensation rate of cloud water and ice (condensation minus evaporation) $Q$ is
calculated following Zhang et al. (2003). In this microphysics scheme, the total grid-scale condensation rates of tagged ice
and tagged cloud water, as well as the total grid-scale evaporation rates of tagged cloud water and tagged ice, are calculated
using the same formulas but the tagged variables are substituted for the corresponding original variables:
$\left(\frac{\partial q_{i,tg}^{k}}{\partial t}\right)_{\text{cond}} = \min\left[A_{\text{st},tg}^{k}A, A_{\text{st},tg}^{k}Q + \frac{q_{i,tg}^{k}}{\Delta t}\right], Q > 0$ (22)
and
$\left(\frac{\partial q_{l,tg}^{k}}{\partial t}\right)_{\text{cond}} = \max\left[A_{\text{st},tg}^{k}Q - \left(\frac{\partial q_{i,tg}^{k}}{\partial t}\right)_{\text{cond}}, 0\right], Q > 0$ (23)
and
$$\left(\frac{\partial q_{l,tg}^k}{\partial t}\right)_{evap} = \max\left(A_{st,tg}^k, -\frac{q_{l,tg}^k}{\Delta t}\right), Q < 0 \qquad (24)$$
and
$$\left(\frac{\partial q_{i,tg}^k}{\partial t}\right)_{evap} = \max\left[A_{st,tg}^k Q - \left(\frac{\partial q_{l,tg}^k}{\partial t}\right)_{evap}, -\frac{q_{i,tg}^k}{\Delta t}\right], Q < 0 \qquad (25)$$
where $A$ is the in-cloud deposition rate of water vapor onto cloud ice (see Eq. (21) of Morrison and Gettelman, 2008).

### 2.4.2 Conversion of cloud water to rain and conversion of cloud ice to snow

The grid-mean autoconversion and accretion rates of water cloud in CAM5.1 are expressed in Eqs. (27) and (28) of Morrison
and Gettelman (2008). Both the two rates can be regard as a term multiply by $A_{st}$. Therefore, the grid-mean autoconversion
and accretion rates of tagged water cloud can be calculated in the same formula but $A_{st,tg}^k$ is substituted for $A_{st}$:
$$\left(\frac{\partial q_{l,tg}^k}{\partial t}\right)_{auto} = \frac{A_{st,tg}^k}{A_{st}}\left(\frac{\partial q_l}{\partial t}\right)_{auto} = -\left(\frac{\partial q_{r,tg}^k}{\partial t}\right)_{auto} \qquad (26)$$
and
$$\left(\frac{\partial q_{l,tg}^k}{\partial t}\right)_{accr} = \frac{A_{st,tg}^k}{A_{st}}\left(\frac{\partial q_l}{\partial t}\right)_{accr} = -\left(\frac{\partial q_{r,tg}^k}{\partial t}\right)_{accw} \qquad (27)$$
where $q_{r,tg}^k$ is the MMR of tagged stratiform rain.
Similarly, the grid-mean autoconversion rate of ice to form snow can be looked as a term multiply by $A_{st}$ (see Eq. (29) of
Morrison and Gettelman (2008)), as well as the accretion of ice followed Lin et al. (1983). Thus, the autoconversion and
accretion rates of tagged ice to form snow are expressed as
$$\left(\frac{\partial q_{i,tg}^k}{\partial t}\right)_{auto} = \frac{A_{st,tg}^k}{A_{st}}\left(\frac{\partial q_i}{\partial t}\right)_{auto} = -\left(\frac{\partial q_{s,tg}^k}{\partial t}\right)_{auto} \qquad (28)$$
and
$$\left(\frac{\partial q_{i,tg}^k}{\partial t}\right)_{accs} = \frac{A_{st,tg}^k}{A_{st}}\left(\frac{\partial q_i}{\partial t}\right)_{accs} = -\left(\frac{\partial q_{s,tg}^k}{\partial t}\right)_{acci} \qquad (29)$$
where $q_{s,tg}^k$ is the MMR of tagged stratiform snow.

### 2.4.3 Other collection processes

The accretion of cloud water by snow $\left(\frac{\partial q_l}{\partial t}\right)_{\text{accs}} = -\left(\frac{\partial q_s}{\partial t}\right)_{\text{accw}}$ is attained by the continuous collection equation, whose
collection efficiency is a function of the Stokes number following Thompson et al. (2004). Similar to the calculation of
$\left(\frac{\partial q_l}{\partial t}\right)_{\text{auto}}$, $\left(\frac{\partial q_l}{\partial t}\right)_{\text{accs}}$ can be regarded as a term multiply by $A_{l,\text{st}}$. Thus, $\left(\frac{\partial q_{l,tg}^k}{\partial t}\right)_{\text{accs}}$ is computed using the same equation but by
multiplying with $A_{l,\text{st},tg}^k$ instead of $A_{l,\text{st}}$:
$$\left(\frac{\partial q_{l,tg}^k}{\partial t}\right)_{\text{accs}} = \frac{A_{l,\text{st},tg}^k}{A_{l,\text{st}}}\left(\frac{\partial q_l}{\partial t}\right)_{\text{accs}} = -\left(\frac{\partial q_{s,tg}^k}{\partial t}\right)_{\text{accw}} \tag{30}$$


The collection of rain by snow $\left(\frac{\partial q_r}{\partial t}\right)_{\text{coll}} = -\left(\frac{\partial q_s}{\partial t}\right)_{\text{coll}}$ can also be regarded as a term multiplied by $A_{\text{st}}$. Therefore,
$\left(\frac{\partial q_{r,tg}^k}{\partial t}\right)_{\text{coll}}$ is computed using the same formula but by multiplying with $A_{\text{st},tg}^k$ instead of $A_{\text{st}}$:
$$\left(\frac{\partial q_{r,tg}^k}{\partial t}\right)_{\text{coll}} = \frac{A_{\text{st},tg}^k}{A_{\text{st}}}\left(\frac{\partial q_r}{\partial t}\right)_{\text{coll}} = -\left(\frac{\partial q_{s,tg}^k}{\partial t}\right)_{\text{coll}} \tag{31}$$


### 2.4.4 Freezing of cloud water and rain

The heterogeneous freezing of cloud water and rain is considered in CAM5.1 (Reisner et al., 1998; Morrison and Pinto,
2005). The heterogeneous freezing of tagged cloud water is computed using the same formula as that of original cloud water,
but by multiplying with $A_{l,\text{st},tg}^k$ instead of $A_{l,\text{st}}$:
$$\left(\frac{\partial q_{l,tg}^k}{\partial t}\right)_{\text{het}} = \frac{A_{l,\text{st},tg}^k}{A_{l,\text{st}}}\left(\frac{\partial q_l}{\partial t}\right)_{\text{het}} \tag{32}$$

Similarly, the heterogeneous freezing of tagged rain is computed using the same formula as that of original rain, but by
multiplying with $A_{\text{st},tg}^k$ instead of $A_{\text{st}}$:
$$\left(\frac{\partial q_{r,tg}^k}{\partial t}\right)_{\text{het}} = \frac{A_{\text{st},tg}^k}{A_{\text{st}}}\left(\frac{\partial q_r}{\partial t}\right)_{\text{het}} \tag{33}$$

The homogeneous freezing of tagged cloud droplets and tagged rain are computed using the same equations as those of the
original cloud droplets and rain, but $q_{l,tg}^k$ and $S_{r,tot,tg}^k$ (the vertical integrated tagged rain source/sink term) are substituted for
the original quantities:
$$\left(\frac{\partial q_{l,tg}^k}{\partial t}\right)_{\text{hom}} = \frac{\left(\frac{\partial q_l}{\partial t}\right)_{\text{hom}}}{\left(\frac{q_l}{\Delta t}\right)}\left(\frac{q_{l,tg}^k}{\Delta t}\right) = -\left(\frac{\partial q_{i,tg}^k}{\partial t}\right)_{\text{hom}} \tag{34}$$
$$\left(\frac{\partial q_{r,tg}^k}{\partial t}\right)_{\text{hom}} = \frac{\left(\frac{\partial q_r}{\partial t}\right)_{\text{hom}}}{S_{r,tot}} S_{r,tot,tg}^k = -\left(\frac{\partial q_{s,tg}^k}{\partial t}\right)_{\text{hom}} \tag{35}$$

### 300 2.4.5 Melting of cloud ice and snow

Similar to the calculations of the homogeneous freezing of cloud water and rain, the melting of tagged ice and tagged snow
are computed using the same equations as those of the original ice and snow, but $q_{i,tg}^k$ and $S_{s,tot,tg}^k$ (the vertical integrated
tagged snow source/sink term) are substituted for the original quantities:
$$\left(\frac{\partial q_{i,tg}^k}{\partial t}\right)_{\text{melt}} = \frac{\left(\frac{\partial q_i}{\partial t}\right)_{\text{melt}}}{\left(\frac{q_i}{\Delta t}\right)}\left(\frac{q_{i,tg}^k}{\Delta t}\right) = -\left(\frac{\partial q_{l,tg}^k}{\partial t}\right)_{\text{melt}} \tag{36}$$
$$\left(\frac{\partial q_{s,tg}^k}{\partial t}\right)_{\text{melt}} = \frac{\left(\frac{\partial q_s}{\partial t}\right)_{\text{melt}}}{S_{s,tot}} S_{s,tot,tg}^k = -\left(\frac{\partial q_{r,tg}^k}{\partial t}\right)_{\text{melt}} \tag{37}$$

### 307 2.4.6 Evaporation/sublimation of precipitation

For the calculations of the evaporation of tagged rain and the sublimation of tagged snow, both them are calculated using the
same formula as original quantities but $A_{\text{st},tg}^k$ is substituted for $A_{\text{st}}$:
$$\left(\frac{\partial q_{r,tg}^k}{\partial t}\right)_{\text{evap}} = \frac{A_{\text{st},tg}^k}{A_{\text{st}}}\left(\frac{\partial q_r}{\partial t}\right)_{\text{evap}} \tag{38}$$
and

$$\left(\frac{\partial q_{s,tg}^k}{\partial t}\right)_{evap} = \frac{A_{st,tg}^k}{A_{st}}\left(\frac{\partial q_s}{\partial t}\right)_{evap} \tag{39}$$


### 2.4.7 Sedimentation of cloud water and ice

The time tendencies ($\left(\frac{\partial q_l}{\partial t}\right)_{sed}$ and $\left(\frac{\partial q_i}{\partial t}\right)_{sed}$) of cloud water and ice for sedimentation, as well as those ($\left(\frac{\partial q_{l,tg}^k}{\partial t}\right)_{sed}$ and
$\left(\frac{\partial q_{i,tg}^k}{\partial t}\right)_{sed}$) of tagged cloud water and tagged ice, are calculated with a simple forward differencing scheme in the vertical
dimension (Morrison and Gettelman, 2008). In CAM5.1, the sedimentation of cloud water and ice can lead to evaporation or
sublimation when the cloud fraction at the level above is larger than the cloud fraction at the given level and the evaporation
or condensation rate is assumed to be proportional to the difference in cloud fraction between the levels. This assumption is
also applied to calculate the evaporation of tagged cloud water or sublimation of tagged ice, when the tagged cloud fraction
at the level above is larger than the tagged cloud fraction at the given level.

### 2.4.8 The diagnosis of precipitation

The grid-scale time tendency of the MMR of precipitation $q_p$ in CAM5.1's microphysics is expressed as:

$$\frac{\partial q_p}{\partial t} = \frac{1}{\rho}\frac{\partial(V_q \rho q_p)}{\partial z} + S_q \tag{40}$$

where $z$ is height, $V_q$ is the mass-weighted terminal fall speeds (see Eq. (18) of Morrison and Gettelman (2008)), and $S_q$ is
the grid-mean source/sink terms for $q_p$:

$$S_q = \left(\frac{\partial q_p}{\partial t}\right)_{auto} + \left(\frac{\partial q_p}{\partial t}\right)_{accw} + \left(\frac{\partial q_p}{\partial t}\right)_{acci} + \left(\frac{\partial q_p}{\partial t}\right)_{het} + \left(\frac{\partial q_p}{\partial t}\right)_{hom} + \left(\frac{\partial q_p}{\partial t}\right)_{melt} + \left(\frac{\partial q_p}{\partial t}\right)_{evap} + \left(\frac{\partial q_p}{\partial t}\right)_{coll} \tag{41}$$

For the diagnostic treatments of tagged rain and tagged snow, the $q_p$ in Eqs. (40) and (41) is replaced by $q_{r,tg}^k$ and $q_{s,tg}^k$,
respectively.

## 2.5 Advection


The finite volume dynamical core is chosen in this study due to its excellent properties for tracer transport (Rasch et al.,

2006). The CAM5.1 model can be driven by offline meteorological fields (Lamarque et al., 2012) following the procedure
initially developed for the Model of Atmospheric Transport and Chemistry (MARCH) (Rasch et al., 1997). This procedure
allows for more accurate comparisons between measurements of atmospheric composition and CAM5.1's output (Lamarque
et al., 2012). In this study, the external meteorological fields are obtained from Modern Era Retrospective-analysis for
Research and Applications (MERRA) datasets (Rienecker et al., 2011), whose horizontal resolution is identical to CAM5.1's
and time resolution is 6 h. In the simulation procedure, the zonal and meridional wind components, air temperature, surface
pressure, surface temperature, surface geopotential, surface stress, and sensible and latent heat fluxes are read from the
MERRA datasets to drive CAM5.1 (Lamarque et al., 2012). To prevent jumps, all input fields are linearly interpolated at
timesteps between the reading times. Later, these fields are used to drive the CAM5.1's parameterizations to generate the
necessary variables and calculate subgrid scale transport and the hydrological cycle (Lamarque et al., 2012). Temporal
evolutions of $q_{v,tg}^k$, $q_{l,tg}^k$, and $q_{i,tg}^k$ in the advective process are treated in the same manner as other constituents without any
modification.

## 2.6 Vertical diffusion

CAM5.1's moist turbulence scheme is taken from the scheme presented by Bretherton and Park (2009), which calculates the
vertical transport of heat, moisture, horizontal momentum, and tracers by symmetric turbulences. The vertical diffusion of
tagged water substances is treated by the procedure in the same way as other constituents without any modification.

## 2.7 Adjustment

Ideally, the differences between the MMRs of water substances and the summed MMRs of all corresponding tagged water substances should be zero. However, there are exceptional differences in a few grid points (see supplementary Fig. S6). Supplementary Figs. S1–S5 show comparisons between the tendencies of the original water substances and the sum of the tendencies of the tagged water substances for the relevant physical processes described in Sects. 2.1 through 2.6. Although differences are small for most grid points, some abnormal values still appear randomly. For tagged water vapour, evident biases mainly occur in deep convection, cloud processes (cloud macrophysics and microphysics), and advection in the tropics; for tagged cloud droplets, the apparent biases generally occur in cloud processes; for tagged cloud ice, the main differences occur in cloud processes, advection, and vertical diffusion. Nonlinearities in the calculations of the tendencies of water substances in the physical schemes cause these differences. A bias occurred in one physical parameterization can affect the calculations of the tendencies of tagged water substances in other parameterizations, since there are interactions among various physical and dynamical processes in CAM5.1. Eventually, clear differences between the summed MMRs of tagged water substances and the MMRs of original water substances may occur, as shown in Fig. S6. To reduce these accumulated biases in the relevant physical schemes, additional criteria are applied to the relevant quantities of the tagged water substances:

(1) If the positive or negative sign of the tendency of a tagged water substance is identical to the sign of the tendency of the original water substance, the absolute value of the tendency of the tagged water substance should not be larger than that of the original water substance. If their signs are different, the tendency of the tagged water substance is set to zero. This adjustment can be expressed as:

$$\frac{\partial q_{tg}^k}{\partial t} = \begin{cases} \max\left(\frac{\partial q_{tg}^k}{\partial t}, \frac{\partial q}{\partial t}\right), & \text{if } \frac{\partial q_{tg}^k}{\partial t} \geq 0 \text{ and } \frac{\partial q}{\partial t} \geq 0 \\ \min\left(\frac{\partial q_{tg}^k}{\partial t}, \frac{\partial q}{\partial t}\right), & \text{if } \frac{\partial q_{tg}^k}{\partial t} \leq 0 \text{ and } \frac{\partial q}{\partial t} < 0 \\ 0, & \text{if } \left(\frac{\partial q_{tg}^k}{\partial t} < 0 \text{ and } \frac{\partial q}{\partial t} \geq 0\right) \text{ or } \left(\frac{\partial q_{tg}^k}{\partial t} > 0 \text{ and } \frac{\partial q}{\partial t} < 0\right) \end{cases} \tag{42}$$

where $\frac{\partial q_{tg}^k}{\partial t}$ and $\frac{\partial q}{\partial t}$ represent the tendency of the tagged water substances and the tendency of the corresponding original water substance in a given physical process, respectively.

(2) After the adjustment in Eq. (42) being applied, the sum of the tendencies of all tagged water substances should be equal
to the tendency of the corresponding original water substance in each scheme. This adjustment can be described as
follows:
$$\frac{\partial q_{tg}^k}{\partial t} = \begin{cases} R_q \left( \frac{\partial q_{tg}^k}{\partial t} \right), & \text{if } \sum_{k=1}^n \left( \frac{\partial q_{tg}^k}{\partial t} \right) \neq 0, \text{ here } R_q = \frac{\frac{\partial q}{\partial t}}{\sum_{k=1}^n \left( \frac{\partial q_{tg}^k}{\partial t} \right)} \\ \frac{1}{n} \left( \frac{\partial q_{tg}^k}{\partial t} \right), & \text{if } \sum_{k=1}^n \left( \frac{\partial q_{tg}^k}{\partial t} \right) = 0. \end{cases}$$
(43)


## 3. Results and discussion

### 3.1 Model assessment

Numaguti (1999) pointed out that the results of the tagged AWTs method suffer from the bias of the model used. Therefore,
we first estimate the precipitation in the specified dynamics simulation of CAM5.1, which is compared to the Global
Precipitation Climatology Project (GPCP) version 2.2 combined precipitation data set (Huffman and Bolvin, 2011), as
shown in Fig. 2. In winter (December, January and February), high-precipitation zones are located in the tropics of the
Southern Hemisphere and in the mid-latitude areas of the NWP. Precipitation is generally less than 3 mm d$^{-1}$ over most parts
of Eurasia. In summer (June, July and August), there is heavy precipitation over the southern and southeastern parts of
Eurasia and over central Africa. Although CAM5.1 generally shows a bias towards relatively high precipitation in the tropics
of the summer hemisphere, the precipitation pattern and amount over Eurasia and its adjacent areas is captured well by
CAM5.1. In addition, the water vapour data from the Atmospheric Infrared Sounder (AIRS) and wind field data from
National Centers for Environmental Prediction (NCEP) are used to assess the CAM5.1's results, as shown in Fig. S7. Overall,
the water vapour and horizontal wind fields can be well simulated by CAM5.1.

## 3.2 Terrestrial and oceanic contributions to precipitation over Eurasia

Figure 3 shows the spatial distribution of the relative contribution of evaporation from all land source regions to precipitation (colours). In winter, evaporation from land source regions generally contributes ~30–60% to the precipitation over Eurasia. The largest contribution (~80%) is located in central China. In summer, ≥60% of precipitation over most parts of Eurasia is supplied by evaporation from land, especially for the inland region where ≥80% of precipitation originates from the land surface. However, the contribution of evaporation from land to summer precipitation over IND, ICP, and east China is generally less than 50%, due to moisture transport by the Indian summer monsoon and EASM. Overall, the contribution of evaporation from land to precipitation over Eurasia is smaller in winter and larger in summer, which is consistent with the variation of evaporation from the land surface over Eurasia in winter and summer as shown in Fig. 4. The pattern of precipitation contributed by land evaporation is similar to that shown in Numaguti (1999). Our result is close to that of Numaguti (1999) for summer but the contribution of land evaporation to precipitation is evidently larger for winter.

The distributions of the relative contributions of evaporation from the NAO, the extended north Indian Ocean (includes NIO, BOB, and AS), and the extended Northwest Pacific (includes NWP and SCS), which are three important moisture source regions, are shown in Fig. 5. In winter, ~10–60% of the precipitation over the northern part of Eurasia originates from the NAO, with a westward or northwestward increasing gradient in the relative contribution. The extended north Indian Ocean supplies moisture for ~10–30% of the precipitation over North Africa and South Asia. The extended Northwest Pacific only provides moisture for 10–30% of the precipitation over the southern and eastern coastal regions of Asia. In summer, evaporation from the NAO only affects precipitation over Europe, with a contribution of 10–30% to total precipitation. Precipitation areas influenced by the extended north Indian Ocean extend to EA, while areas impacted by the extended Northwest Pacific retreat eastward.

The arrow streamlines in Fig. 3 show the total tropospheric water vapour flux in winter and summer. There is a westward component of water vapour flux over the tropics of both the extended north Indian Ocean and the extended Northwest Pacific in the Northern Hemisphere in winter. In summer, there is a very large northwestward water vapour flux over the

NIO, turning northeastward over the BOB and AS. Over the extended Northwest Pacific, there is a northward component of
water vapour flux at 30 °–60 °N and a westward flux in the tropics between 120 °E and 180 °E. In addition, Fig. 4 shows strong
surface evaporation over the NWP and NAO in winter, while evaporation is weaker in summer. In contrast, evaporation over
the NIO is larger in summer and smaller in winter. These results help to explain the variations in the contributions of the
NAO, extended north Indian Ocean, and extended Northwest Pacific to precipitation in winter and summer as shown in Fig.

423  5.


The overall contributions from these three oceanic regions are generally less than those in Numaguti (1999). The resolution
of the climate model used in Numaguti (1999) is ~5.6 °, both in latitudinal and longitudinal direction. The different model
resolutions are a probable reason for the different quantitative contributions in our study and that of Numaguti (1999). In
addition, CAM5.1 is driven by MERRA data, so its surface evaporation flux is approximate to that of MERRA. MERRA
land evaporation is larger over South and East Asia and Northern Europe compared to other global estimates (Jiménez et al.,
2011), and Bosilovich et al. (2011) suggested that MERRA ocean evaporation is lower compared to other reanalyses but is
much closer to observation. Therefore, the bias in MERRA surface evaporation may lead to the higher land contribution and
lower oceanic contribution to precipitation.

**3.3 Atmospheric moisture source attribution of precipitation and water vapour over the YRV**
Figures 6a and 6b show the time series of evaporative contribution of each source region to precipitation over the YRV. The
contributions of evaporation to precipitation from the BOB and AS are lower during autumn–winter and higher during
spring–summer with relative contributions of ≤3.9%. Chow et al. (2008) (see their Fig. 20a) also found that evaporation from
the AS had little impact on precipitation over China. Supplementary Figs. S10–S13 show the distributions of 25 tagged water
vapour tracers and 25 tagged precipitations over Eurasia and surrounding areas in winter and summer. Figs. S10a and S12a
show that evaporation from the BOB contributes to water vapour and precipitation over the extended north Indian Ocean in
winter, corresponding to the direction of water flux shown in Fig. 3a. The centre of BOB-contributed precipitation (15 mg m$^-$
$^2$ s$^{-1}$) is located in the south of the TP in summer (Fig. S13a). In addition, the BOB supplies moisture to areas around the
northeastern BOB in summer (Fig. S11a). The contribution of the SCS to precipitation is also very small (≤3.4%), which
supports the view of Chow et al. (2008), who suggested that the SCS may serve as a pathway for water vapour transport
from the southwesterly flow of the Indian summer monsoon and the easterly flow of the Northwest Pacific subtropical high.
A detailed discussion of this issue is presented in Sect. 3.5. The NWP serves as the dominant oceanic source region for
precipitation over the YRV during the whole year except during June and July. The relative contribution is ~8.1–10.6% in
June and July and 15.8–24.6% in other months. As shown in Fig.3, there is strong westward water vapour flux over 20°–45°
N for the NWP and southwestward water vapour flux over the tropics of the NWP. However, there is no evident moisture
transports from the NWP to EA in the long term mean water vapour flux. Following Eq. (S1), the water vapour flux is
divided into the stationary and transient components, as shown in Figs. S8–S9. The transient component of the meridional
flux brings some of the moisture from south over most of the NWP and the north of the SCS (Fig. S8c), and the transient
component of the zonal flux leads to westwards water vapour transport over 20°–30°N for the NWP (Fig. S9c). Both the
transient components indicate that the synoptic disturbances can bring moisture originating from the NWP to the southern
and eastern coastal regions of Asia during winter. Evaporation from the NIO shows a clear contribution to precipitation
during May to October. In particular, the NIO is the dominant oceanic source region in June and July, with a contribution of
~30%. This is in agreement with the result of a Lagrangian diagnostic method described in Baker et al. (2015) and the results
of sensitivity experiments in Chow et al. (2008). However, in other months, the contribution of the NIO is very small. The
contributions from evaporation from the BOB, AS, and NIO are in phase with the EASM, which was also reported by Baker
et al. (2015). The ICP is an important terrestrial source region for the YRV precipitation, supplying moisture to ~9.9% of the
annual precipitation. The relative contribution of the ICP from April to September is close to the result of Wei et al. (2012).
The contribution of evaporation from the YRV to its precipitation can be regarded as the local recycling ratio, which is lower
(4.5–7.4%) in summer and higher (9.2–13.4%) in other seasons. In general, the contribution of evaporation from SCN is
comparable to the local contribution of the YRV. The relative contribution from the NEA is higher in autumn–winter and
lower in spring–summer, which may be associated with the shift of the EA monsoon. Though the individual contributions of
evaporation from the YRV or SCN are smaller than those from the NIO in summer, their combined contributions exceed
10%. This implies that evaporation from these two regions is important for precipitation over China. This is contrary to the
view expressed in Simmonds et al. (1999) and Qian et al. (2004), but consistent with Wei et al. (2012). Figures 6c and 6d
show a time series of evaporative contribution from each source region to the tropospheric water vapour amount over the
YRV. The overall relative contribution from each source region to the total water vapour amount is similar to the
corresponding relative contribution to precipitation shown in Figs. 6a and 6b.

**3.4 Atmospheric moisture source attribution of precipitation and water vapour over SCN**
Figures 7a and 7b show the contribution of each source region to precipitation over SCN. The NIO is the dominant source
region in summer, while the NWP dominates precipitation over SCN during other seasons, which is similar to the situation
over the YRV. The contribution from the NIO is 28.4–37.8% in summer. The contribution from the NWP is 8.7–17.2% in
summer and ~15.3–37.2% during other seasons. During spring and summer, ~2–4.4% of precipitation is supplied from the
BOB, with smaller contributions during other seasons. The contribution from the AS is similar to that of the BOB. In
summer, only 2.7–3.7% of precipitation originates from the SCS, but the area contributes ~6.7–7% to the precipitation in
early spring (March to April). Similar to precipitation over the YRV, the dominant terrestrial source region for SCN is the
ICP, which contributes ~9.8% to the precipitation. In addition, ~5.6% of summer precipitation originates from SEA.
Compared to precipitation over the YRV, the contribution from the TP is smaller. In addition, the contribution from the YRV
is small in summer. The local recycling ratio or percentage contribution of evaporation from SCN is generally 4.3–7.2%
during May to September, but larger than 9.3% during the remaining months. As shown in Fig. 7d, the overall relative
contribution of each source region to the water vapour amount is similar to each region's contribution to precipitation over
SCN.

## 3.5 Atmospheric moisture source attribution of water vapour over the SCS

Simmonds et al. (1999) and Lau et al. (2002) suggested that interannual variation of summer precipitation over China is associated with water vapour transport over the SCS. However, Chow et al. (2008) suggested that the SCS may act as a water vapour transport pathway where the southwesterly stream of the Indian summer monsoon and the easterly stream of the southeastern Asian monsoon meet. Previous studies have conducted sensitivity experiments or analysed the water vapour budget to indirectly determine moisture sources for the SCS. In contrast, our AWT method can directly quantify the contribution of each source region to the water vapour amount over the SCS, which is shown in Fig. 8. The local contribution of the SCS is small (~4.7–5.5%) in summer, and the mean contribution in other months is ~6.8%. The contribution of the NIO shows clear seasonal variations: the contribution is high during May to October, but very small during the other months. Similar to the results for water vapour over the YRV and SCN, the NIO is the dominant source region from June to September, with a contribution of 22.7–31%. During this period, the contribution of the NWP is 14.1–21.2%. However, the NWP dominates the water vapour over the SCS in the remaining months, with contributions of 25.7–51.3%. In addition, the SP and NEP are also important oceanic source regions, with combined annual contributions of ~11–16.6%. The most important terrestrial moisture source region is the SEA, whose contribution is larger (13.8–16.2%) in summer and smaller (~5.3%) in winter. During late autumn to winter, about 5.3–6.3% of water vapour is supplied from NEA, but its contribution is very small in other seasons. The other land source regions contribute relatively little to the water vapour amount over the SCS.

From the SCS to SCN and further to the YRV (from south to north), surface evaporation from the SCS generally represents a small (≤5.5%) contribution to the water vapour amount over the three target areas in summer. In contrast, much more water vapour is supplied by evaporation from the NWP and NIO. This confirms the inference proposed by Chow et al. (2008) that the SCS is a water vapour transport pathway where moisture from the NIO and NWP meet in summer.

## 4. Conclusions

In this study, an Eulerian tagged AWT method was implemented in CAM5.1, which provides the capacity to separately trace the behaviour of atmospheric water substances originating from various moisture source regions and to quantify their contributions to atmospheric water over an arbitrary region. Numaguti (1999) pointed out that the weakness of the tagged AWT method is that its results suffer from the performance of the model in reproducing the hydrological cycle. However, a comparison between GPCP and CAM5.1 precipitation shows that CAM5.1 has the capability to represent total precipitation. CAM5.1 also can reproduce water vapour and large scale circulation reasonable, as compared to AIRS and NCEP data. Using this method, we investigated the contribution of evaporation from land, as well as the contributions from the North Atlantic Ocean, extended north Indian Ocean, and extended Northwest Pacific to precipitation over Eurasia. Our results are similar to those of Numaguti (1999), except that our results indicate a larger contribution from terrestrial source regions, while the three oceanic regions show smaller contributions. Different model resolutions and a bias in MERRA surface evaporation are probable causes for the differences between our results and those of Numaguti (1999).

We then investigated the contribution of various source regions to precipitation and water vapour amounts over the YRV and SCN. Our results suggest that the dominant oceanic moisture source region during summer is the NIO (20.5–30.3% of precipitation over the YRV; 28.4–37.8% of precipitation over SCN), consistent with Baker et al. (2015) and Chow et al. (2008), while during other seasons, the NWP is the dominant source region (15.8–24.6% of precipitation over the YRV; 15.3–37.1% of precipitation over SCN), with smaller contributions from the BOB, AS, and SCS. The ICP is an important terrestrial source region, with a mean annual contribution of ~10%. For precipitation over the YRV, the combined contribution of evaporation from the YRV and SCN is non-negligible (exceeding 10%), consistent with Wei et al. (2012). For precipitation over SCN, the local recycling ratio is generally 4.3–7.2% during May to September, and reaches 9.4–18.7% in other months. The contribution from the YRV is very small in summer. The overall relative contribution of each source region to the water vapour amount is similar to the corresponding contribution to precipitation over the YRV and SCN.

An analysis of water vapour amount over the SCS shows that the NIO is the dominant source region (22.7–31% of water
vapour) during June to September, while the NWP dominates (25.7–51.3% of water vapour) in the remaining months. In
contrast, the local contribution of the SCS is smaller (~4.7–5.5%) in summer. In addition, the SP, NEP, and SEA are also
important source regions. Evaporation over the SCS represents a small contribution to water vapour amounts over the SCS,
SCN, and the YRV in summer, implying that the SCS acts as a water vapour transport pathway rather than a dominant
source region, which confirms the inference of Chow et al. (2008).

At present, the tagged AWT method has only been applied to a few GCMs and regional models, and has generally focused
on identifying the moisture distribution over a few regions such as North America (Bosilovich and Schubert, 2002;
Bosilovich et al., 2003). We expect that the AWT method will be applied to additional models and used to identify moisture
sources over more climate regions, which will improve our understanding of atmospheric moisture transport.

**Code availability**
The source code modifications for CAM5.1 are available from the authors. Interested readers should contact us via
arthur_pc@163.com or binzhu@nuist.edu.cn.

**Acknowledgements:**
This work is supported by grants from the National Natural Science Foundation of China (Grant No. 91544229), the
National Key Research and Development Program of China (2016YFA0602003), and the projects of China Special Fund for
Meteorological Research in the Public Interest (GYHY201406001).

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

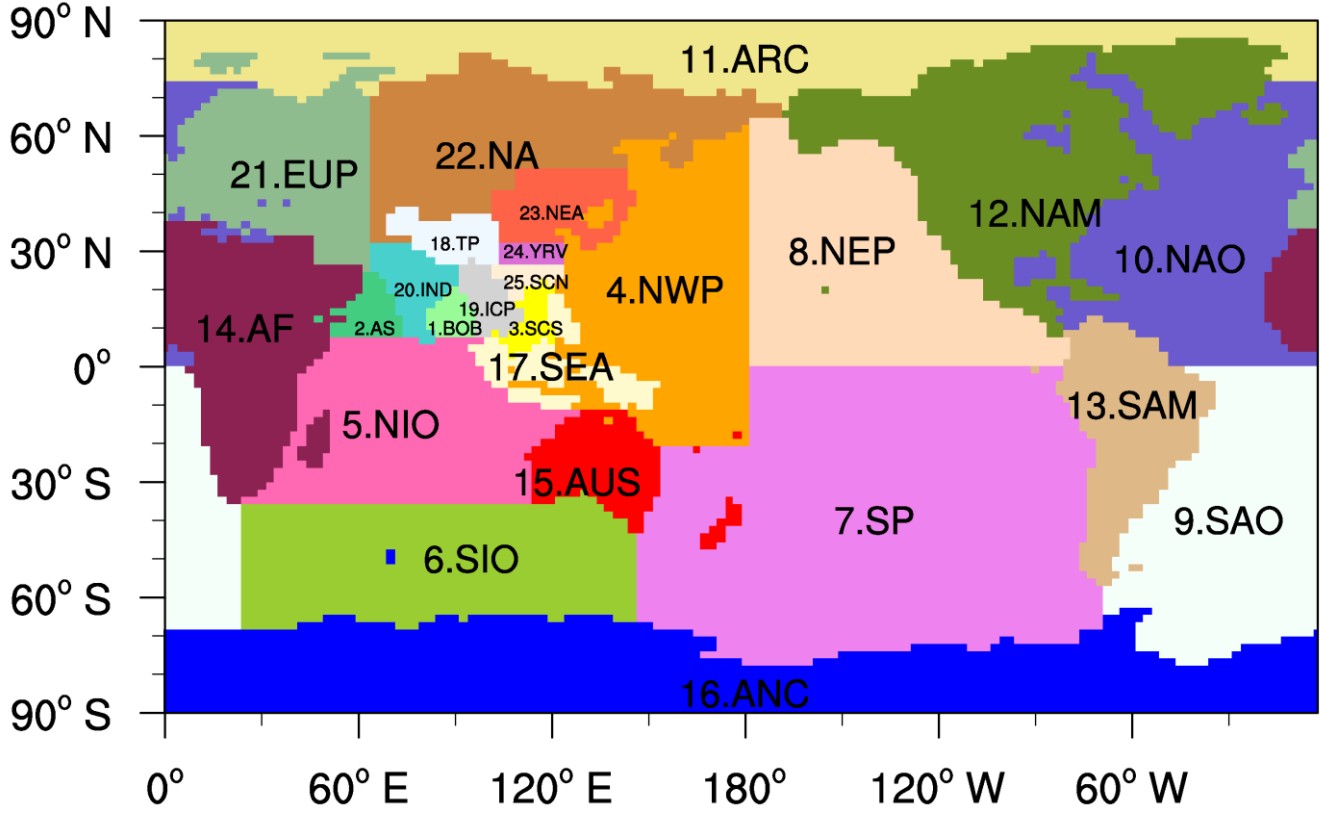

**Figure 1.** Moisture source regions: the regions are denoted as (1) Bay of Bengal: BOB; (2) Arabian Sea: AS; (3) South China Sea: SCS; (4) Northwest Pacific: NWP; (5) north Indian Ocean: NIO; (6) southern Indian Ocean: SIO; (7) southern Pacific: SP; (8) Northeast Pacific: NEP; (9) southern Atlantic Ocean: SAO; (10) northern Atlantic Ocean: NAO; (11) Arctic Ocean: ARC; (12) North America: NAM; (13) South America: SAM; (14) Africa: AF; (15) Australia: AUS; (16) Antarctic: ANC; (17) Southeast Asia: SEA; (18) Tibet Plateau: TP; (19) Indo-China Peninsula: ICP; (20) India: IND; (21) Europe: EUP; (22) North Asia: NA; (23) Northeast Asia: NEA; (24) Yangtze River Valley: YRV; (25) South China: SCN.

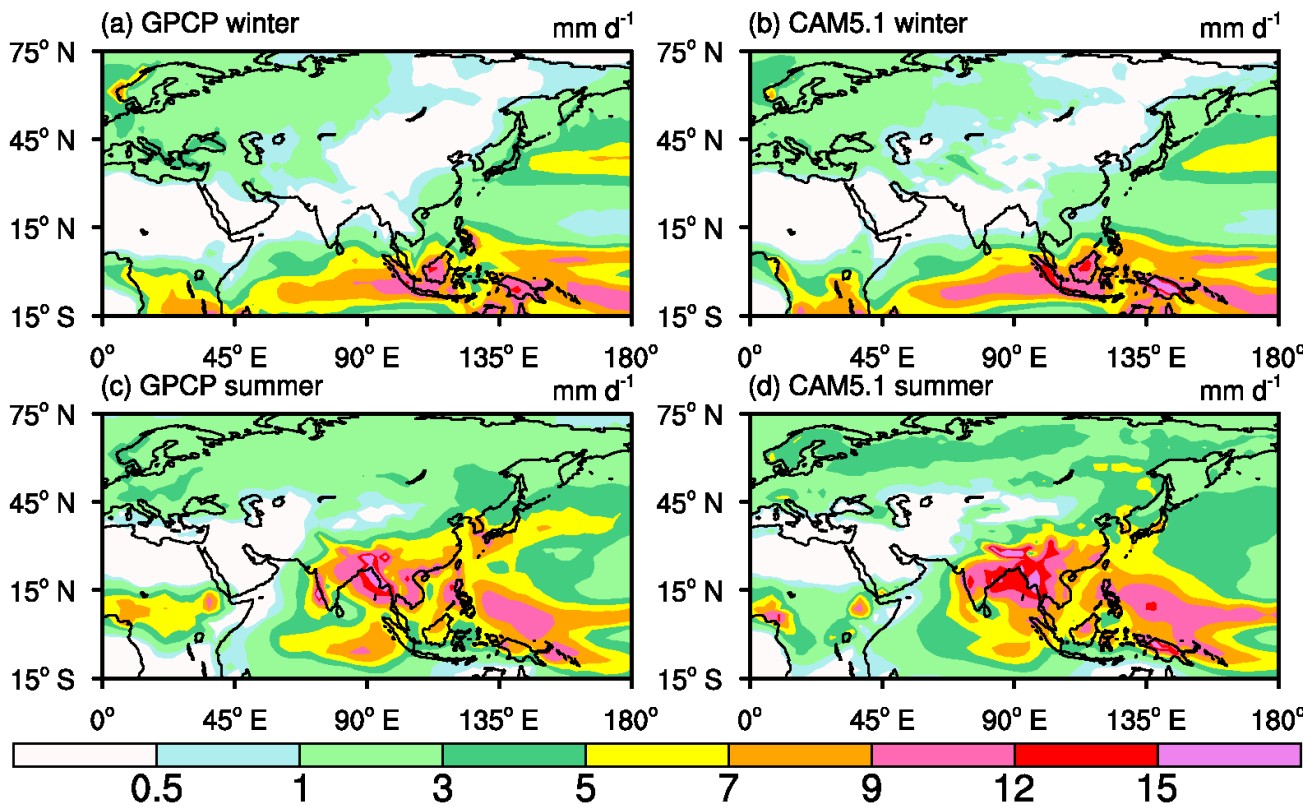


**Figure 2.** Comparisons between (left) GPCP data and (right) CAM5.1 precipitation simulations during (top) winter and (bottom) summer

(ten-year averages for 1998 to 2007).

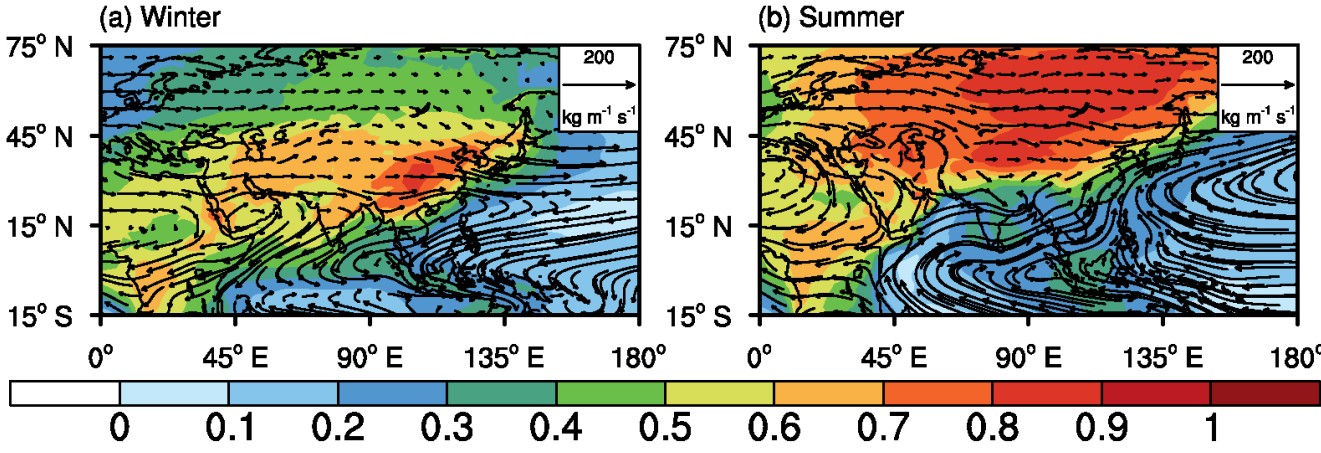


**Figure 3.** Distribution of the relative contribution to precipitation from all land source regions defined in **Fig. 1** (colours, unit: ratio of
tagged precipitation over total precipitation) and the vertically integrated total tropospheric water vapour flux (arrow streamlines, unit: kg
$m^{-1}$ $s^{-1}$) during (a) winter and (b) summer.

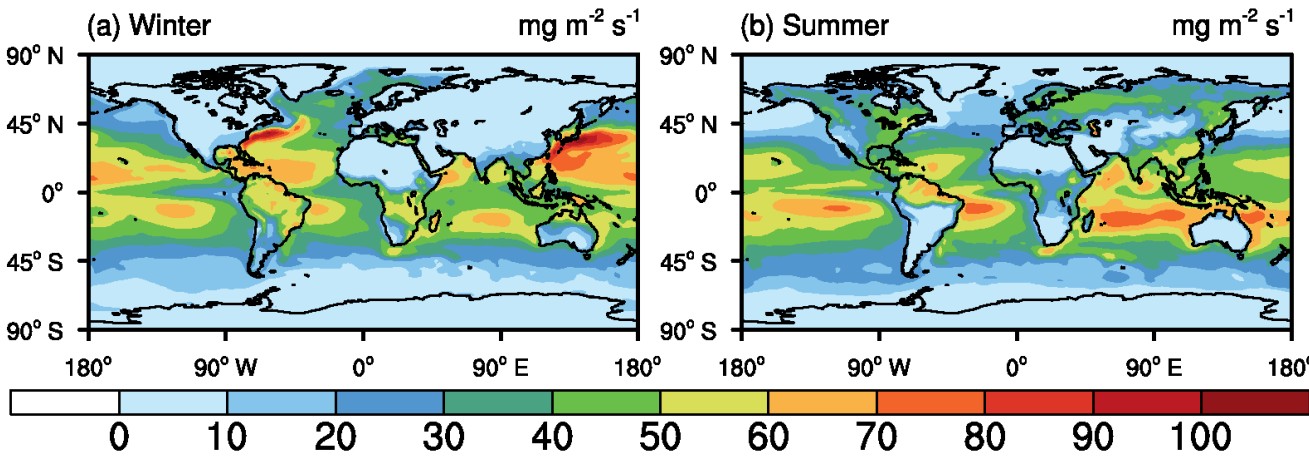

Figure 4. Distribution of CAM5.1's ten-year averaged surface evaporation flux (unit: mg m$^{-2}$ s$^{-1}$) in (a) winter and (b) summer between 1998 and 2007.

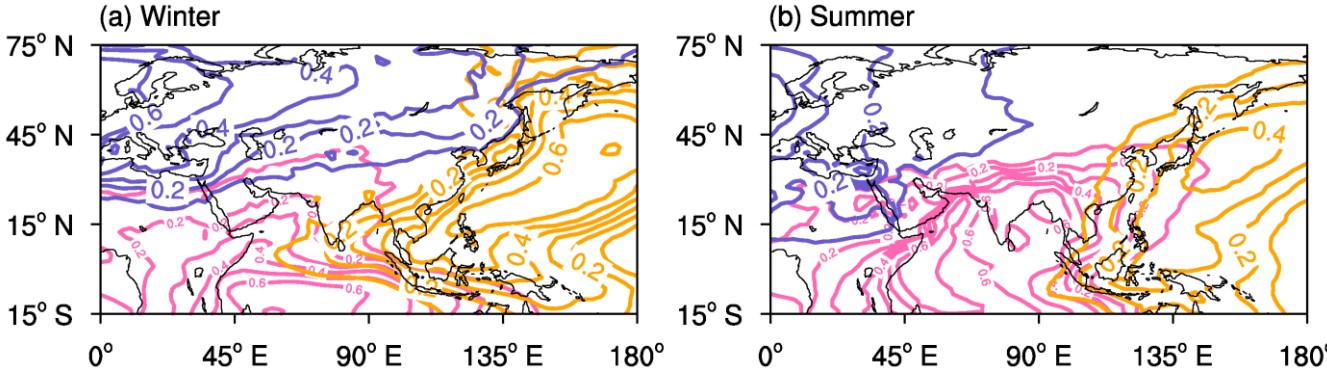


**Figure 5.** Distributions of the ratios of precipitation (unit: ratio of tagged precipitation over total precipitation) supplied from the NAO
(slate blue), the extended north Indian Ocean (NIO + BOB + AS, pink), and the extended Northwest Pacific (NWP + SCS, orange) during
(a) winter and (b) summer. Contour interval is 0.1.

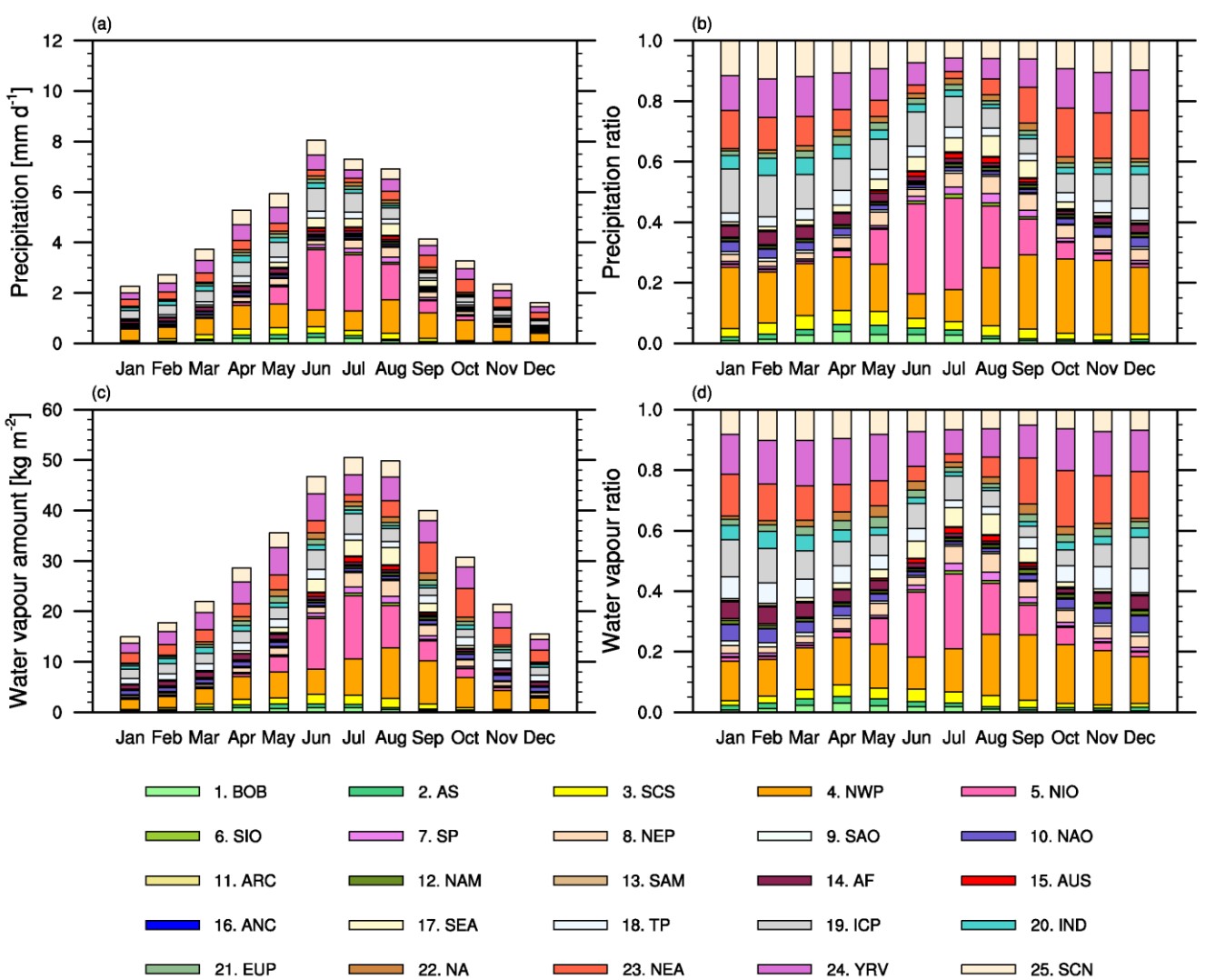

**Figure 6.** (a) Monthly averaged evaporative contributions of 25 defined source regions to the precipitation over the YRV. (b) Same as **Fig. 6a**, but for the relative contribution to precipitation. (c) Monthly averaged evaporative contributions of 25 defined source regions to the tropospheric total water vapour amount over the YRV. (d) Same as **Fig. 6c**, but for the relative contribution to water vapour. Stacked column colours correspond to source region colours in **Fig. 1**.

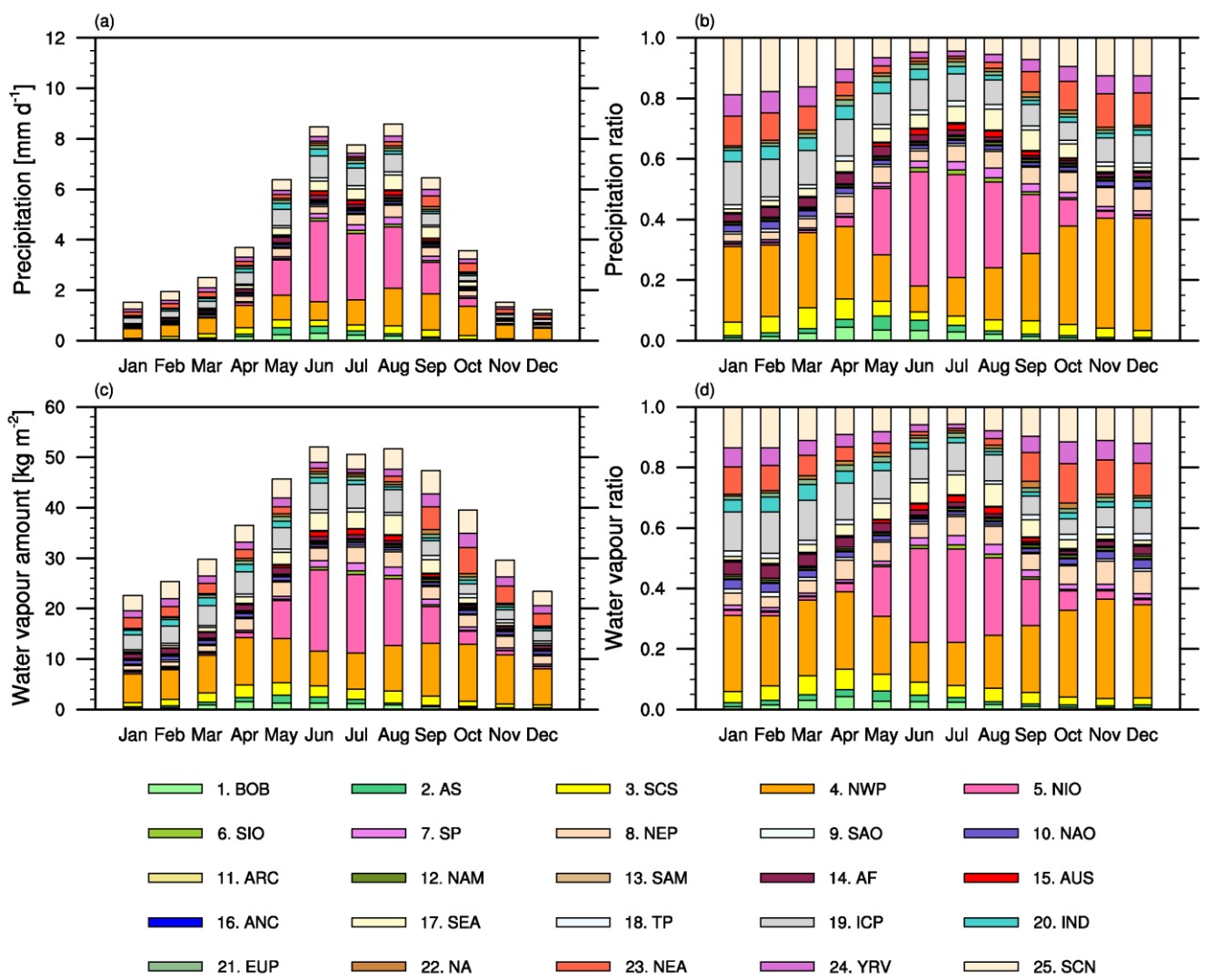


**Figure 7.** Same as **Fig. 6**, but for the contributions and relative contributions of 25 source regions to precipitation and tropospheric total
water vapour amount over SCN.

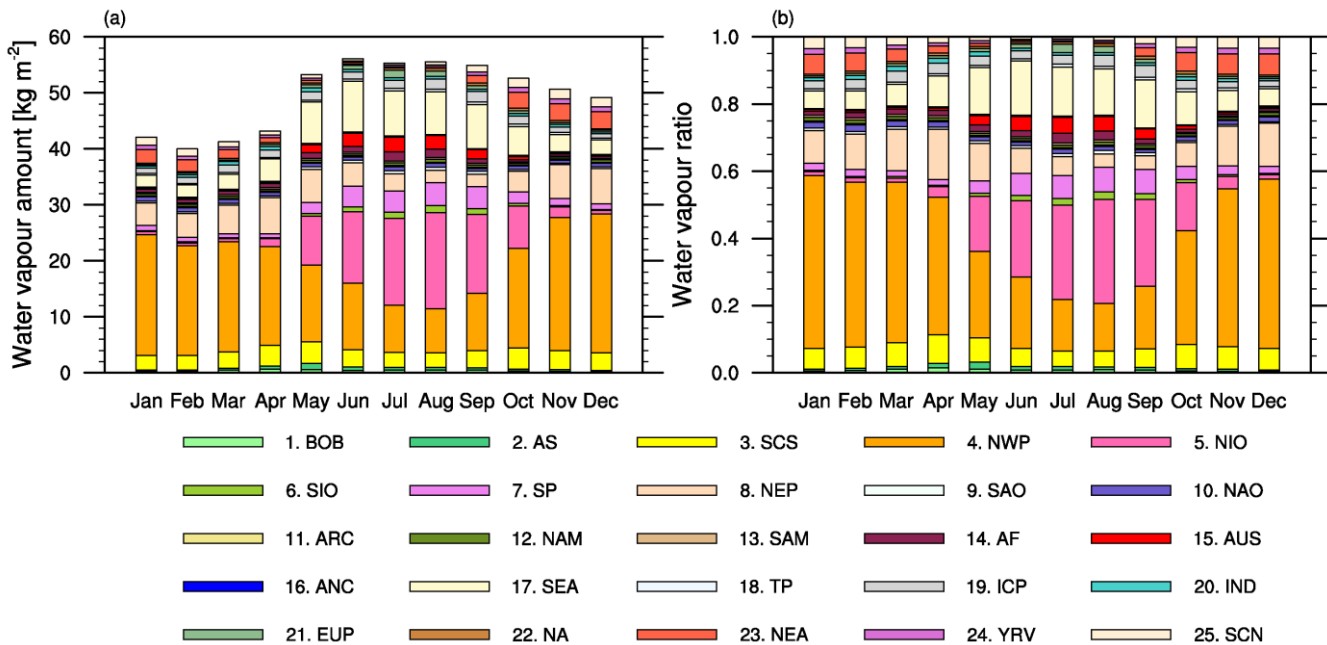


731 **Figure 8.** (a) Monthly averaged evolution of evaporative contribution of 25 defined source regions to the tropospheric total water vapour

732 amount over the SCS. (b) Same as **Fig. 8a**, but for the relative contribution of water vapour. Stacked column colours correspond to source

733 region colours in **Fig. 1**.