# Peer review of "Source apportionment of atmospheric water over East Asia – a source tracer study in CAM5.1"

_Geoscientific Model Development, 2016_

## Referee Comment (RC1) · Anonymous Referee #1 · 18 Oct 2016

**Review of "Source apportionment of atmospheric water over East Asia – a source tracer study in CAM5.1" for *Geoscientific Model Development*.**

**Summary:**

This paper describes the implementation of a new Atmospheric Water Tracer (AWT) scheme in the NCAR Community Atmosphere Model Version 5.1 (CAM5.1). This new feature is then used to examine the sources of precipitation and water vapor for the Yangtze River Valley, Southern China, and the South China Sea. It is found that the North Atlantic, Northwest Pacific, and Northern Indian Ocean are the dominant moisture sources for the three regions, along with evaporation from Asia itself. In particular, it is found that the Indian ocean-based moisture sources tended to be largest in summer, during the monsoons, while the Pacific was the largest source during the rest of the year.

**Recommendation:**

The application of atmospheric water tracers to the Southeast Asian region is certainly interesting, and can provide new insights into the hydrological cycle and processes of this important region. However, there is potentially one major flaw in the implementation of the water tracers in CAM5.1 that must be addressed before it is fully accepted, as well as a few other concerns that are listed in the next few sections. Thus I am recommending **major revisions** for this article. Once these issues have been dealt with, then I believe the paper will be ready for publication.

**Major issues:**

1. This is not an issue in terms of the science presented here, but it is important to note that water tracers already exist in CAM5, up through CAM5.3, and is at least partially described here:

http://onlinelibrary.wiley.com/doi/10.1002/2016MS000649/full

Thus although this does not take away from the science results presented here, it might be more beneficial if this work was presented in a less model-development focused journal, as these particular developments have already been done for this same model.

2. If am understanding your description of the water tracer implementation correctly, then the way you are treating the water tag tendencies from deep convection is sadly not valid, and will cause mass conservation issues which could put into question the scientific results shown here. The reason is because the convective tendency is partly generated by the transport of water vapor in the vertical, which may not have the same water tracer ratio as the level at which you are calculating the tendency. Thus this change in the ratio will result in the implicit addition or removal of water mass. For example, one component of the deep convective vapor tendency is:

$$\frac{\partial q_v}{\partial t} = \frac{\partial}{\partial z}(M_u q_u)$$

Using your formulation, the resulting tag equation would be:

$$\frac{\partial q_{v,tg}^k}{\partial t} = R\frac{\partial q_v}{\partial t} = R\left(\frac{\partial}{\partial z}(M_u q_u)\right), \qquad R = \frac{q_{v,tg}^k}{\sum_{k=1}^n q_{v,tg}^k}$$

Discretizing the vertical derivative results in something akin to:

$$\frac{\partial q_{v,tg}^k}{\partial t} = R\left(\frac{(M_{z_2} q_{z_2} - M_{z_1} q_{z_1})}{z_2 - z_1}\right) = \frac{(M_{z_2} R q_{z_2} - M_{z_1} R q_{z_1})}{z_2 - z_1}$$

Which shows that the only way your formulation can work is if the water tag ratios were exactly the same for both vertical levels on which the deep convection is being applied, which is almost certainly not true. Otherwise, the assumed ratio will be different than the actual water tag ratio, and thus result in a mass conservation error.

The only way to eliminate this problem is to have the water tracer tendency calculated in the exact same way as regular water, e.g.:

$$\frac{\partial q_{v,tg}^k}{\partial t} = E_{tg}^k - C_{tg}^k - \frac{1}{\rho}\frac{\partial}{\partial z}\left(M_u q_{u,tg}^k + M_d q_{d,tg}^k - M_c q_{v,tg}^k\right)$$

where the phase changes are calculated using the ratio method you described:

$$C_{tg}^k = \left(\frac{q_{v,tg}^k}{\sum_{k=1}^n q_{v,tg}^k}\right) C$$

If this is how you are actually doing it, then I would recommend just re-wording this section. However, if this is an issue, then I must recommend you either modify your existing algorithms to fix this issue, or simply re-do your experiments with the already existing water tracer implementation present in CAM5. Finally, I should note that the reason this error may not be showing up in your supplemental error figures is because you are examining the sum of all your water tags, and not the individual tags themselves (and thus allowing mass conservation errors of different signs to cancel each other out).

**Minor issues:**

1. On line 27, I would avoid stating that water vapour is the most important component of the atmosphere, as that is probably just one's opinion. Instead maybe say something like "water vapour is one of the most important components of the atmosphere".

2. It is unclear to me how you are calculating the water tracer vapor tendency produced by the shallow convection, as most of the description focuses solely on the condensate. Could you add a sentence or two describing the shallow convection's water tracer vapor tendency? Also, if it is implemented in the same way as the deep convection, then the major issue described above will also need to be dealt with for the shallow convection as well.

3. In Section 2.7, it is stated that the sum of all tagged water tendencies should be equal to the tendency of the standard water model substance. However, it is unclear what occurs if this rule is

violated.  In particular, if this requirement is not met, what is done to the individual water tracers themselves in order to ensure that the summed tendencies are brought back to the value of the standard water tendency?

4.  One line 537, I would replace "(colours, unit:  1)" with "(colours, unit: ratio of tagged precip over total precip)", or at least something that is more descriptive than just the number one.  I would do the same for the "unit: 1" reference in Figure 5 as well.

5.  I would describe what the vectors are in the caption of Figure 3 as is done in the main text.

6.  In Figure 5b, it is difficult to tell which pink contour corresponds to the "0.2" amount, as the label overlaps multiple contour lines.  If possible, can you shift the label over such that it is more clear which contour line it is referring to?  Possibly making the label smaller might also help in this situation as well.

7.  It might be good to include some sort of legend for Figures 6, 7, and 8 that re-states which water tag each color corresponds to.  This will help lessen the reader's need to constantly go back to Figure 1 to determine what water tag each color represents.

**Grammatical issues:**

1.  Need to make sure that when you have list of three or more objects in a sentence, that commas are used like so:

x, y, and z

Instead, you often times have:

x, y and z

This makes it seem that y and z are together as one idea, when in reality they are separate.  So, just make sure to have a comma before the "and" whenever a list is involved.

2.  On lines 81 and 82, replace "isotope data not only reflect the water cycle" with "isotope data reflects more than just the water cycle".  The reason being that "not only" is a conjunction I believe, and so the phrase would need a ",but" at the end, as in "not only x, but also y".

3.  On line 82, I would replace "and that sensitivity" with just "and sensitivity".

4.  On lines 97 and 99, replace "Neal" with "Neale".

5.  On line 222, I would replace "sum MMRs" with "summed MMRs".

6.  On line 241, I would replace "compared with" with "compared to".

7.  On line 266, I would replace "over the North Africa" with just "over North Africa".

8.  On line 366, I would replace "NAO" with "North Atlantic Ocean", as none of the other regional acronyms are used in this particular sentence.

9.  On line 390, I would replace "over few regions" with "over a few regions".  It might also be beneficial to spell out NAM instead of using the acronym here, although that is probably just personal preference.

---

## Referee Comment (RC2) · Anonymous Referee #2 · 7 Nov 2016

Review comments on "Source apportionment of atmospheric water over East Asia – a source tracer study in CAM5.1" by Pan et al.

This study uses CAM5.1 to identify the sources of moisture contributing to the precipitation in East Asia. Both approach and results are interesting. The manuscript may be accepted for publication in GMD after major revision. Specific comments are listed below.

Major comments:

I.    Diagnostics part:

  1.   It is clear how the simulation were conducted. The simulation were conducted from 1997-2007. It is said in line 284 that "…CAM5.1 is driven by MERRA data …". Obviously, it was not an AMIP-type experiment. Please provide clear discussion the simulation procedure and how the MERRA data were applied to drive CAM5.1. Also what does "offline version of CAM5.1" mean?

  2.   In addition to comparing the simulated precipitation with GPCC, a comparison of simulated water substances and convective/stratiform precipitation with satellite observation could be useful and informative. Also, assessments on other parts of water cycle, such as the evaporation, surface water storage, and their seasonal cycles (e.g. Numaguti 1999) should also be checked. The bias of model simulated large-scale circulation and their possible impacts on the results should also be discussed.

  3.   In the simulation, WNP contribution in terms of percentage to YRV precipitation was the largest in cool season. This is not obvious when looking at long-term mean water moisture flux shown in Figure 3. The contribution is likely associated with the synoptic disturbances that could bring moisture from south. The authors may need to provide their views somewhere in the text. Moisture transport is likely contributed by a large portion by synoptic disturbances. But the manuscript tends to discuss the related dynamics based only on long-term mean water vapor flux.

  4.   Contribution from each region is difficult to distinguish in the bar charts shown in Figure 6-8. Could authors re-plot bar charts by stacking all regions according to their region number (e.g., 1 to 25 from bottom to top) and present a schematic showing the stacking scheme?

II.   Tagged AWTs

1. The approaches for adding tagged water vapor and qc and qi within individual physical parameterizations need more detailed description in section 2, especially for the macrophysics and microphysics schemes. For example in macrophysics, I does not quite understand how the tagging of those microphysical, advection, and convective tendency from other processes in solving Park's matrix in the macrophysics was done. Similarly, details for those complicated microphysical processes were not discussed. Also, snow and rain (important sink of tagged water) were diagnostically determined in the microphysics of CAM5.1 version. Snow and rain are important sinks of tagged water. However, no discussion on these two hydrometeors was provided.

2. How were the detrained qc and qi from deep and shallow convection schemes tagged and put into macrophysics?

3. How was the adjustment exactly done when the sum of tendencies of all tagged water substances was not equal to the tendency of the corresponding original substance? How big the adjustment can be? Would the results be quite different if no adjustment were done?

4. In CAM5, evaporation of convective rainfall is assumed to be as Sunqvist (1988), which is proportional to the square root of the total rainwater flux at each level. Therefore, the linear partitioning of evaporation based on precipitation flux of tagged water (eq.2) does not seem to be consistent with the formulation used in the model.

5. Some of the formulation of tagged water substance in the macrophysics are confusing, especially for the cloud fraction. How can the stratus cloud fraction be composed proportionally of each tagged condensates without mixing?

Minor comments:
1. It is "Neale et al." rather than "Neal et al." in the text (line 99).
2. It is "Gettelman" rather than "Gettleman" in the text (line 189, 190, 196).
3. It is not clear what Figure S7-S10 exactly show. To where and what the vapour tracers supplied from 25 source regions contribute?

---

## Author Comment (AC1) · 30 Dec 2016

Please see supplemental files for response to referees, the mark-up version of the revised manuscript, and the supplement of the revised manuscript.

Please also note the supplement to this comment:
http://www.geosci-model-dev-discuss.net/gmd-2016-255/gmd-2016-255-AC1-supplement.zip
* * *

---

## Author Response (AR1)

**Response to the referees**

Dear editor and referees,

We sincerely appreciate the time and attention you have devoted to our manuscript. Your comments and suggestions were very helpful in improving our manuscript. We have responded to all the referees' comments regarding our manuscript titled "Source apportionment of atmospheric water over East Asia – a source tracer study in CAM5.1". Please find our responses below:

**Referee #1:**

**Summary:**

**This paper describes the implementation of a new Atmospheric Water Tracer (AWT) scheme in the NCAR Community Atmosphere Model Version 5.1 (CAM5.1). This new feature is then used to examine the sources of precipitation and water vapor for the Yangtze River Valley, Southern China, and the South China Sea. It is found that the North Atlantic, Northwest Pacific, and Northern Indian Ocean are the dominant moisture sources for the three regions, along with evaporation from Asia itself. In particular, it is found that the Indian ocean-based moisture sources tended to be largest in summer, during the monsoons, while the Pacific was the largest source during the rest of the year.**

**Recommendation:**

**The application of atmospheric water tracers to the Southeast Asian region is certainly interesting, and can provide new insights into the hydrological cycle and processes of this important region. However, there is potentially one major flaw in the implementation of the water tracers in CAM5.1 that must be addressed before it is fully accepted, as well as a few other concerns that are listed in the next few sections. Thus I am recommending major revisions for this article. Once these issues have been dealt with, then I believe the paper will be ready for publication.**

**Major issues:**

**1. This is not an issue in terms of the science presented here, but it is important to note that water tracers already exist in CAM5, up through CAM5.3, and is at least partially described here: http://onlinelibrary.wiley.com/doi/10.1002/2016MS000649/full**

**Thus although this does not take away from the science results presented here, it might be more beneficial if this work was presented in a less model-development focused journal, as these particular developments have already been done for this same model.**

Reply: The atmospheric water tracers (AWTs) method implemented in CAM5.1 in our manuscript is entirely developed by ourselves. The AWTs method in our manuscript enables the CAM5 model to quantitatively trace the behaviour of atmospheric water substances originating from their moisture source region. We believe that our method is a technical improvement to the CAM5.1 model and is of interest to the readership of the GMD journal.

The method used in Singh et al. (2016) provides a similar feature regarding the atmospheric water tagging problem, although there may be some differences in the treatments of the relevant water tagging calculations (the details of the water tracer method are partially described in their paper). We believe that the two methods improve the CAM5 model system. We are also very pleased to communicate with Singh and other researchers through which we can further improve the water tracer method.

**2. If am understanding your description of the water tracer implementation correctly, then the way you are treating the water tag tendencies from deep convection is sadly not valid, and will cause mass conservation issues which could put into question the scientific results shown here. The reason is because the convective tendency is partly generated by the transport of water vapor in the vertical, which may not have the same water tracer ratio as the level at which you are calculating the tendency. Thus this change in the ratio will result in the implicit addition or removal of water mass. For example, one component of the deep convective vapor tendency is:**

$$\frac{\partial q_v}{\partial t} = \frac{\partial}{\partial z}(M_u q_u)$$

**Using your formulation, the resulting tag equation would be:**

$$\frac{\partial q_{v,tg}^k}{\partial t} = R\frac{\partial q_v}{\partial t} = R\left(\frac{\partial}{\partial z}(M_u q_u)\right), \qquad R = \frac{q_{v,tg}^k}{\sum_{k=1}^n q_{v,tg}^k}$$

**Discretizing the vertical derivative results in something akin to:**

$$\frac{\partial q_{v,tg}^k}{\partial t} = R\left(\frac{(M_{z_2}q_{z_2} - M_{z_1}q_{z_1})}{z_2 - z_1}\right) = \frac{(M_{z_2}Rq_{z_2} - M_{z_1}Rq_{z_1})}{z_2 - z_1}$$

**Which shows that the only way your formulation can work is if the water tag ratios were exactly the same for both vertical levels on which the deep convection is being applied, which is almost certainly not true. Otherwise, the**

assumed ratio will be different than the actual water tag ratio, and thus result in a mass conservation error. The only way to eliminate this problem is to have the water tracer tendency calculated in the exact same way as regular water, e.g.:

$$\frac{\partial q_{v,tg}^k}{\partial t} = E_{tg}^k - C_{tg}^k - \frac{1}{\rho}\frac{\partial}{\partial z}\left(M_u q_{u,tg}^k + M_d q_{d,tg}^k - M_c q_{v,tg}^k\right)$$

where the phase changes are calculated using the ratio method you described:

$$C_{tg}^k = \left(\frac{q_{v,tg}^k}{\sum_{k=1}^{n} q_{v,tg}^k}\right) C$$

If this is how you are actually doing it, then I would recommend just re-wording this section. However, if this is an issue, then I must recommend you either modify your existing algorithms to fix this issue, or simply re-do your experiments with the already existing water tracer implementation present in CAM5. Finally, I should note that the reason this error may not be showing up in your supplemental error figures is because you are examining the sum of all your water tags, and not the individual tags themselves (and thus allowing mass conservation errors of different signs to cancel each other out).

Reply: Thank you for your suggestion. This is an important improvement to our study. In the revised manuscript, we have modified the algorithms to fix this issue following your suggestion. Further, we performed our experiments again and we have added new results in the revised manuscript. Though there are some changes in numerical results, the qualitative conclusions remain unchanged.

Changes in manuscript: Please see Sect. 2.1 in the revised manuscript, which describe the tagged water vapour tendency in the deep convection scheme. Please also see the new numerical results in abstract and Sects. 3.3–4 in the revised manuscript.

**Minor issues:**

**1. On line 27, I would avoid stating that water vapour is the most important component of the atmosphere, as that is probably just one's opinion. Instead maybe say something like "water vapour is one of the most important components of the atmosphere".**

Reply: Thank you for your suggestion. We have revised this sentence following your suggestion in the revised manuscript.

**2. It is unclear to me how you are calculating the water tracer vapor tendency produced by the shallow convection, as most of the description focuses solely on the condensate. Could you add a sentence or two describing the shallow convection's water tracer vapor tendency? Also, if it is implemented in the same way as the deep convection, then the major issue described above will also need to be dealt with for the shallow convection as well.**

Reply: Thank you for your suggestion. The implementation of the shallow convection's water tracer vapour tendency is different from that of the deep convection. The mass mixing ratio (MMR) of the total water is assumed to be a conserved quantity in non-precipitating moist adiabatic processes within the shallow convection scheme. This assumption is also applied to the MMR of the tagged total water. Therefore, the tendency of tagged water vapour is computed as the difference between the tendency of tagged total water and the tendencies of tagged condensates in non-precipitating processes within the shallow convection scheme, similar to the calculation of the tendency of water vapour.

Changes in manuscript: We have added similar sentences to describe the shallow convection's tagged water vapour tendency in Sect. 2.2 in the revised manuscript.

**3. In Section 2.7, it is stated that the sum of all tagged water tendencies should be equal to the tendency of the standard water model substance. However, it is unclear what occurs if this rule is violated. In particular, if this requirement is not met, what is done to the individual water tracers themselves in order to ensure that the summed tendencies are brought back to the value of the standard water tendency?**

Reply: (a) If the summed tendencies of the tagged water substances are not equal to the tendencies of the corresponding original water substances in one physical parameterization, calculations of the tendencies of tagged water substances in other parameterizations may be affected as there are interactions among various physical and dynamical processes in CAM5.1. Clear differences between the summed MMRs of tagged water substances and the MMRs of original water substances may occur as shown in Fig. S6. We have added several sentences in Sect. 2.7 in the revised manuscript to state what may occur if the rule is violated.

(b) We have rewritten the description of the adjustment criteria, ensuring that the summed tendencies are returned to the value of the standard water tendency. Equations (42) and (43) have also been added to express these adjustments. Please see Sect. 2.7 in the revised manuscript, in which we have implemented these changes.

**4. One line 537, I would replace "(colours, unit: 1)" with "(colours, unit: ratio of tagged precip over total precip)", or at least something that is more descriptive than just the number one. I would do the same for the "unit: 1" reference in Figure 5 as well.**

Reply: Thank you for your suggestion. We have replaced "(colours, unit: 1)" with "(colours, unit: ratio of tagged precipitation over total precipitation)" in the captions of Figure 3 and Figure 5 in the revised manuscript.

**5. I would describe what the vectors are in the caption of Figure 3 as is done in the main text.**

Reply: Thank you for your suggestion. The vectors have been described in the caption of Figure 3 in the revised manuscript.

**6. In Figure 5b, it is difficult to tell which pink contour corresponds to the "0.2" amount, as the label overlaps multiple contour lines. If possible, can you shift the label over such that it is more clear which contour line it is referring to? Possibly making the label smaller might also help in this situation as well.**

Reply: Thank you for your suggestion. We have decreased the size of the label and increased the density of the label for the pink contour in Figure 5 in the revised manuscript.

**7. It might be good to include some sort of legend for Figures 6, 7, and 8 that re-states which water tag each color corresponds to. This will help lessen the reader's need to constantly go back to Figure 1 to determine what water tag each color represents.**

Reply: Thank you for your suggestion. We have added legends for Figures 6, 7, and 8 in the revised manuscript.

**Grammatical issues:**

**1. Need to make sure that when you have list of three or more objects in a sentence, that commas are used like so: x, y, and z Instead, you often times have: x, y and z This makes it seem that y and z are together as one idea, when in reality they are separate. So, just make sure to have a comma before the "and" whenever a list is involved.**

Reply: Thank you for your suggestion. We have checked our manuscript carefully and modified the sentences with the aforementioned problem in the revised manuscript.

**2. On lines 81 and 82, replace "isotope data not only reflect the water cycle" with "isotope data reflects more than just the water cycle". The reason being that "not only" is a conjunction I believe, and so the phrase would need a ",but" at the end, as in "not only x, but also y".**

Reply: Thank you for your suggestion. We have replaced "isotope data not only reflect the water cycle" with "isotope data reflects more than just the water cycle" in Sect. 1 in the revised manuscript.

**3. On line 82, I would replace "and that sensitivity" with just "and sensitivity".**

Reply: Thank you for your suggestion. We have replaced "and that sensitivity" with just "and sensitivity" in the revised manuscript.

**4. On lines 97 and 99, replace "Neal" with "Neale".**

Reply: Thank you for your suggestion. We have replaced "Neal" with "Neale" in the revised manuscript.

**5. On line 222, I would replace "sum MMRs" with "summed MMRs".**

Reply: Thank you for your suggestion. We have replaced "sum MMRs" with "summed MMRs" in Sect. 2.7 in the revised manuscript.

**6. On line 241, I would replace "compared with" with "compared to".**

Reply: Thank you for your suggestion. We have replaced "compared with" with "compared to" in Sect. 3.1 in the revised manuscript.

**7. On line 266, I would replace "over the North Africa" with just "over North Africa".**

Reply: Thank you for your suggestion. We have replaced "over the North Africa" with "over North Africa" in the second paragraph of Sect. 3.2 in the revised manuscript.

**8. On line 366, I would replace "NAO" with "North Atlantic Ocean", as none of the other regional acronyms are used in this particular sentence.**

Reply: Thank you for your suggestion. We have replaced "NAO" with "North Atlantic Ocean" in the first paragraph of Sect. 4 in the revised manuscript.

**9. On line 390, I would replace "over few regions" with "over a few regions". It might also be beneficial to spell out NAM instead of using the acronym here, although that is probably just personal preference.**

Reply: Thank you for your suggestion. We have replaced "over few regions" with "over a few regions". "NAM" has also been replaced by "North America". Please check them in the last paragraph of Sect. 4 in the revised manuscript.

**Referee 2#:**

This study uses CAM5.1 to identify the sources of moisture contributing to the precipitation in East Asia. Both approach and results are interesting. The manuscript may be accepted for publication in GMD after major revision. Specific comments are listed below.

**Major comments:**

**I. Diagnostics part:**

**1. It is clear how the simulation were conducted. The simulation were conducted from 1997-2007. It is said in line 284 that "…CAM5.1 is driven by MERRA data …". Obviously, it was not an AMIP-type experiment. Please provide clear discussion the simulation procedure and how the MERRA data were applied to drive CAM5.1. Also what does "offline version of CAM5.1" mean?**

Reply: Thank you for your suggestion.

(a) The basic simulation setup is identical to that in Lamarque et al. (2012). We have cited the work of Lamarque et al. (2012) instead of restating it again. Please see the second paragraph in Sect. 2 in the revised manuscript: "The basic simulations setup, including emissions and upper and lower boundary conditions, is identical to that of the specified dynamics simulations of CAM5 in Lamarque et al. (2012). In this study, the wet removal scheme in Horowitz et al. (2003) is adopted."

(b) The zonal and meridional wind components, air temperature, surface pressure, surface temperature, surface geopotential, surface stress, and sensible and latent heat fluxes are read from the MERRA datasets to drive CAM5.1. All input fields are linearly interpolated at timesteps between the reading times to prevent jumps. Subsequently, these fields are used to drive the CAM5.1's parameterizations to generate the necessary variables and calculate subgrid scale transport and the hydrological cycle (Lamarque et al., 2012). We have provided these descriptions of how the MERRA data were applied to drive CAM5.1. Please check it in Sect. 2.5 in the revised manuscript.

(c) "Offline version of CAM5.1" means that the CAM5.1 model is driven by external meteorological fields. In the revised manuscript, we have replaced "precipitation simulated by the offline version of CAM5.1" with "precipitation in the specified dynamics simulation of CAM5.1" in Sect. 3.1 in the revised manuscript.

**2. In addition to comparing the simulated precipitation with GPCC, a comparison of simulated water substances and convective/stratiform precipitation with satellite observation could be useful and informative. Also, assessments on other parts of water cycle, such as the evaporation, surface water storage, and their seasonal cycles (e.g. Numaguti**

**1999) should also be checked. The bias of model simulated large-scale circulation and their possible impacts on the results should also be discussed.**

Reply: Thank you for your suggestion.

(a) In the revised manuscript, we have added the comparison between the Atmospheric Infrared Sounder (AIRS) observed and CAM5.1 simulated water vapour for the years from 2003 to 2007. In general, the simulated water vapour is well consistent with measured values. Please see Sect. 3.1 in the revised manuscript and Fig. S7 in the supplementary information. In addition, the corresponding conclusion has also been added in Sect. 4 in the revised manuscript.

(b) We compared the cloud water content from AIRS and CAM5.1 simulation results. Please see Fig. R1 below. Overall the cloud water pattern and amount over oceans around Eurasia can be characterized by CAM5.1, but there are some shifts for the high cloud water centres (e.g. over Northwest Pacific) in the model.

[Figure]

Figure R1. Comparison between (left) AIRS observed and (right) CAM5.1 simulated cloud water content (unit: kg m$^{-2}$) during (top) winter and (bottom) summer. All results are 5-year averages for 2003–2007. Grey areas indicate where required data are not available.

(c) Figure R2 shows the comparison between the Microwave Limber Sounder (MLS) observed and simulated cloud ice content at 215 hPa. There is significant lower bias in the simulated result. Waliser et al. (2009) pointed out that the accurate simulation of cloud ice in global circulation models (GCMs) is still a challenge to model development. Because the poor representation of cloud ice in the model, the source apportionment of cloud ice in CAM5.1 cannot provide valid results. However, the total precipitation and water vapour can be reproduced well by CAM5.1; therefore results and discussions in this manuscript focused on the source apportionments of total precipitation and water vapour.

[Figure]

Figure R2. Comparison between (left) MLS observed and (right) CAM5.1 simulated cloud ice content (unit: mg m⁻³) at 215 hPa during (top) winter and (bottom) summer. All results are 3-year averages for 2005–2007. Grey areas indicate where required data are not available.

(d)    The comparisons of simulated convective/stratiform precipitation with the Tropical Rainfall Measuring Mission (TRMM) precipitation radar data are shown in Fig. R3. Convective and stratiform precipitations have comparable magnitudes in TRMM measurement, while the precipitation in CAM5.1 is mainly contributed by the convective precipitation parameterization (Yang et al., 2013). The partition of convective and stratiform precipitations is one of the major challenges in the current model physics (Arakawa, 2004). Dai (2006) also reported that most of the GCMs still have problems in reproducing an accurate magnitude of stratiform precipitation compared to TRMM data. However, because the CAM5.1 model can reproduce the total precipitation reasonably well, we focused on the result from the source apportionment of total precipitation in this study.

[Figure]

Figure R3. Comparison between (top) TRMM observed and (bottom) CAM5.1 simulated convective and stratiform precipitations (unit: mm d$^{-1}$) during (leftmost two columns) winter and (rightmost two columns) summer for 1998–2007.

(e)    We compared the evaporation flux in CAM5.1 with data from the Objectively Analyzed Air-sea Fluxes (OAFlux) Project, as shown in Fig. R4. Overall the evaporation in CAM5.1 is in good agreement with that in OAFlux datasets. In this study, CAM5.1 is driven by MERRA data, in which the latent heat flux is computed from the evaporation flux. Meanwhile, the latent heat flux is used to calculate the surface evaporation in CAM5.1. Therefore, the evaporation flux in CAM5.1 is very close to that in MERRA. Jiménez et al. (2011) and Bosilovich et al. (2011) provided a detailed assessment on the evaporation in MERRA compared to other global estimates and observation. Thus, we cited the two papers instead of re-assessing the evaporation flux again in our manuscript. In general, the evaporation of MERRA over land is larger than the evaporation in other global estimates (Jiménez et al., 2011). Bosilovich et al. (2011) pointed out that the evaporation in MERRA is lower compared to OAFlux, while other estimates generally overestimate evaporation over oceans. In contrast, the MERRA evaporation is much closer to OAFlux data than other estimates. These biases in MERRA data may lead to the higher land contribution and lower oceanic contribution to precipitation in the results from source apportionment in this study. In the revised manuscript, we have mentioned the above discussions in the last paragraph of Sect. 3.2.

[Figure]

Figure R4. Distributions of the evaporation fluxes (unit: mg m$^{-2}$ s$^{-1}$) in OAFlux datasets and CAM5.1 during winter and summer for 1998–2007. Grey areas indicate where required data are not available.

(f)  Here, we compared the simulated terrestrial water storage with Gravity Recovery and Climate Experiment (GRACE) observation, as shown in Fig. R5. We used the three latest land water solutions provided by GFZ, JPL, and CSR in the GRACE data. To be consistent with GRACE observation, the baseline average over 2004 to 2009 are removed in simulated results, as shown in Fig. R5a. The overall seasonal cycle of total water storage can be characterized by the model, but the amplitude in model is significantly larger than that in observations. Note that the evaporation in MERRA is used to drive the CAM5.1 model. Thus, there is no virtual moisture transmit from the Earth's surface into the atmosphere in the specified dynamics simulation of CAM5.1.

[Figure]

Figure R5. (a) The evolution of GRACE-derived anomaly in terrestrial water storage and that in simulated terrestrial water storage relative to the average for 2003–2009 over the land areas within 10 °N–40 °N and 60 °E–120 °E. (b) The evolution of simulated terrestrial water storage.

(g)   In the revised manuscript, we have added a discussion on the assessment of the simulated horizontal wind fields. In general, the horizontal wind fields in CAM5.1 are in agreement with those from National Centers for Environmental Prediction (NCEP). Please see Sect. 3.1 in the revised manuscript and Fig. S7 in the supplementary information. A corresponding conclusion has been added in Sect. 4 in the revised manuscript.

**3. In the simulation, WNP contribution in terms of percentage to YRV precipitation was the largest in cool season. This is not obvious when looking at long-term mean water moisture flux shown in Figure 3. The contribution is likely associated with the synoptic disturbances that could bring moisture from south. The authors may need to provide their views somewhere in the text. Moisture transport is likely contributed by a large portion by synoptic disturbances. But the manuscript tends to discuss the related dynamics based only on long-term mean water vapor flux.**

Reply: Thank you for your suggestion. In the revised manuscript, we divided the meridional and zonal water vapour flux into stationary and transient terms. Please see supplementary Figs. S8–S9. Figure S8c shows that the transient component of the meridional flux brings some of the moisture from south over most of the NWP and the north of the SCS. Figure S9c shows that the transient component of the zonal flux leads to westwards water vapour transport over 20 °–30 °N for the NWP. Both the transient components indicated that the synoptic disturbances could bring moisture originating from the NWP to the southern and eastern coastal regions of Asia during winter. We have added this view in Sect. 3.3 in the revised manuscript.

**4. Contribution from each region is difficult to distinguish in the bar charts shown in Figure 6-8. Could authors re-plot bar charts by stacking all regions according to their region number (e.g., 1 to 25 from bottom to top) and present a schematic showing the stacking scheme?**

Reply: Thank you for your suggestion. For Figs. 6–8, all bar charts are plotted by stacking all regions according to their region number (1 to 25 from bottom to top). We have also added legends to show the stacking scheme in the revised manuscript.

**II. Tagged AWTs**

**1. The approaches for adding tagged water vapor and qc and qi within individual physical parameterizations need more detailed description in section 2, especially for the macrophysics and microphysics schemes. For example in macrophysics, I does not quite understand how the tagging of those microphysical, advection, and convective tendency from other processes in solving Park's matrix in the macrophysics was done. Similarly, details for those complicated microphysical processes were not discussed. Also, snow and rain (important sink of tagged water) were diagnostically determined in the microphysics of CAM5.1 version. Snow and rain are important sinks of tagged water. However, no discussion on these two hydrometeors was provided.**

Reply: Thank you for your suggestion.

(a)  We have added more detailed descriptions on the tagged AWTs method in Sect. 2 in the revised manuscript.

(b)  In the cloud macrophysics, Park et al. (2014) defined the grid-mean net condensation rate of water vapour into liquid stratus condensate $\bar{Q}_l$ as the time change of $\bar{q}_{l,a}$ minus the external forcing (all processes except stratus macrophysics, including stratus microphysics, moisture turbulence, advection, and convection) of cloud droplets $\bar{F}_l$:

$$\bar{Q}_l = \dot{\bar{q}}_{l,a} - \bar{F}_l = A_{l,\text{st}}\dot{q}_{l,\text{st}} + \alpha q_{l,\text{st}}\dot{A}_{l,\text{st}} - \bar{F}_l \tag{R1}$$

where $\dot{\bar{q}}_{l,a}$, $\dot{q}_{l,\text{st}}$, and $\dot{A}_{l,\text{st}}$ are the time tendency of $\bar{q}_{l,a}$, $q_{l,\text{st}}$, and $A_{l,\text{st}}$ during $\Delta t = 1800$ s, respectively. In CAM5.1, $\alpha = 0.1$ is the ratio of newly formed or dissipated stratus to the preexisting $q_{l,\text{st}}$. Similarly, the tagged grid-mean net condensation rate $\bar{Q}_{l,tg}^k$ is calculated as:

$$\bar{Q}_{l,tg}^k = \dot{\bar{q}}_{l,a,tg}^k - \bar{F}_{l,tg}^k = A_{l,\text{st},tg}^k \dot{q}_{l,\text{st}} + \alpha q_{l,\text{st}}\left(R\dot{A}_{l,\text{st}} + A_{l,\text{st}}\dot{R}\right) - \bar{F}_{l,tg}^k, \text{and } R = \frac{\bar{q}_{v,tg}^k}{\sum_{k=1}^{n}\bar{q}_{v,tg}^k} \tag{R2}$$

Here, $\dot{R}$ is the tendency of $R$ during $\Delta t$, and $\bar{F}_{l,tg}^{k}$ is the changes of tagged cloud droplets in processes such as microphysics, moisture turbulence, advection, and deep and shallow convections. We have added these sentences in the last paragraph of Sect. 2.3 in the revised manuscript.

(c)  We have added descriptions on the calculations of tagged water substances in the microphysical processes. Please see Sect. 2.4.1–2.4.7 in the revised manuscript.

(d)  We have added some descriptions on the calculations of tagged snow and tagged rain in the microphysical processes. Please see Sect. 2.4.8 in the revised manuscript.

**2. How were the detrained qc and qi from deep and shallow convection schemes tagged and put into macrophysics?**

**Reply: Thank you for your question.**

(a)  In the deep convection scheme, only the detrainment of cloud water is taken into consideration. Equation (6c) of Zhang and McFarlane (1995) is used to calculate the MMR of cloud water $q_l$ in the updraft. Finally, the detrainment of cloud water is calculated as the product of $q_l$ and the detrainment rate. Similarly, the MMR of tagged cloud water $q_{l,tg}^{k}$ is calculated in the similar equation as the Eq. (6c) of Zhang and McFarlane (1995), but $q_{l,tg}^{k}$ is substituted for $q_l$. The detrainment of tagged cloud water is calculated as the product of $q_{l,tg}^{k}$ and the detrainment rate as well. We have stated that the calculation of the detrainment of tagged cloud water is identical to the detrainment of cloud water, but $q_{l,tg}^{k}$ is substituted for $q_l$. A similar sentence has been added in Sect. 2.1 in the revised manuscript.

(b)  In the shallow scheme, because the detrainment of cloud water and ice ($D(q_l)$ and $D(q_i)$) is assumed to be proportional to the total water detrainment and the detrained air is assumed to be a representative of cumulus updraft (Park and Bretherton, 2009), we use the ratio of tagged total water in the updraft $q_{t,u,tg}^{k}$ and the corresponding sum to distribute the detrainment of tagged cloud water and ice ($D\left(q_{l,tg}^{k}\right)$ and $D\left(q_{i,tg}^{k}\right)$). We have explained this in Sect. 2.2 in the revised manuscript. The corresponding calculation of $q_{t,u,tg}^{k}$ has also been added in Sect. 2.2 in the revised manuscript.

(c)  In the macrophysics, the detrainments of cumulus condensates were added to $q_l$ and $q_i$, and then to compute the final equilibrium state in-stratus. Similarly, the tendencies of detrained cumulus condensates were also added, and Eqs. (13)–(19) in the revised manuscript are used to partition the equilibrium state of tagged water in-stratus.

**3. How was the adjustment exactly done when the sum of tendencies of all tagged water substances was not equal to the tendency of the corresponding original substance? How big the adjustment can be? Would the results be quite different if no adjustment were done?**

Reply: Thank you for your suggestion.

(a)  We have rewritten the description of the adjustment. Equations (42) and (43) have also been added to express the adjustment in the revised manuscript. Please see Sect. 2.7 in the revised manuscript.

(b) The adjustment was used to ensure that the summed MMRs of the tagged water is brought back to the value of the standard water. We have evaluated the differences between the results with adjustment and which without adjustment. The adjustment can cause the change of 4.0–4.8 g m$^{-2}$ for summed tagged water vapour (accounts for 0.02–0.026% of the MMR of water vapour), the change of 0.075–0.11 g m$^{-2}$ for summed tagged cloud water (accounts for 0.14–0.19% of the MMR of cloud water), and the change of 0.15–0.57 g m$^{-2}$ for summed tagged cloud ice (accounts for 2.8–7.7% of the MMR of cloud ice) at the global scale.

(c) If no adjustment was made, the results generally have no significant difference. As shown in Figs. R6–R8, for most of source regions, their water tracers in the calculation with the adjustment are very close to the results from the calculation without the adjustment at the global scale. For source regions such as ANC, IND, and NEA, there are differences in their contributions to water substances between the results with adjustment and which without adjustment. However, contributions from these source regions to water substances are generally small at the global scale. The adjustment has tiny effect on the result to determine the dominant source regions of atmospheric water substances.

[Figure]

Figure R6. Comparisons between the tagged water vapour contents (unit: kg m$^{-2}$) in which the adjustment is applied (black) and the corresponding results with no adjustment (red) for February 1997 to January 1998. Figs. R6(a)–R6(x) correspond to the tagged water vapour originating from the source regions 1–25 defined in Fig. 1, respectively. All results are global average values.

[Figure]

Figure R7. Same as Fig. R6, but for the tagged cloud droplets contents (unit: g m$^{-2}$).

[Figure]

Figure R8. Same as Fig. R6, but for tagged cloud ice contents (unit: $10^{-4}$ kg m$^{-2}$).

**4. In CAM5, evaporation of convective rainfall is assumed to be as Sunqvist (1988), which is proportional to the square root of the total rainwater flux at each level. Therefore, the linear partitioning of evaporation based on precipitation flux of tagged water (eq.2) does not seem to be consistent with the formulation used in the model.**

Reply: The evaporation rate $\left(\frac{\partial q_v^k}{\partial t}\right)_{\text{dp\_evap}}$ at level $m$ is associated with the deep convection precipitation flux $(Q_m)_{\text{dp}}$ at the top interface of this level (Sundqvist, 1998), expressed as

$$\left(\frac{\partial q_v^k}{\partial t}\right)_{\text{dp\_evap}} = k_e(1 - \text{RH}_m)\sqrt{(Q_m)_{\text{dp}}} \tag{R3}$$

where $\text{RH}_m$ is the relative humidity at level $m$ and the coefficient $k_e = 2 \times 10^{-6}$ (kg m$^{-2}$ s$^{-1}$)$^{-1/2}$ s$^{-1}$. The basic idea of the AWT method is to separate the contribution from each source region to the content of atmospheric water substances in each relevant physical process. If Eq. (R3) is used to compute the evaporation rate of tagged convection precipitation

$\left(\frac{\partial q_{v,tg}^k}{\partial t}\right)_{\text{dp\_evap}}$ at level $m$, in most cases $\sum_{k=1}^{n}\left(\frac{\partial q_{v,tg}^k}{\partial t}\right)_{\text{dp\_evap}} \neq k_e(1 - \text{RH}_m)\sqrt{\sum_{k=1}^{n}\left(Q_{m,tg}^k\right)_{\text{dp}}}$ due to the nonlinearity of Eq.

(R3). However, Eq. (R3) is equivalent to

$$\left(\frac{\partial q_v^k}{\partial t}\right)_{\text{dp\_evap}} = k_e(1 - \text{RH}_m)\frac{(Q_m)_{\text{dp}}}{\sqrt{(Q_m)_{\text{dp}}}}, \text{if } (Q_m)_{\text{dp}} \neq 0 \tag{R4}$$

Thus, if $\sum_{k=1}^{n}\left(Q_{m,tg}^k\right)_{\text{dp}} \neq 0$, the summed evaporation rate of tagged precipitation must satisfy the below equation:

$$\sum_{k=1}^{n}\left(\frac{\partial q_{v,tg}^k}{\partial t}\right)_{\text{dp\_evap}} = k_e(1 - \text{RH}_m)\frac{\sum_{k=1}^{n}\left(Q_{m,tg}^k\right)_{\text{dp}}}{\sqrt{\sum_{k=1}^{n}\left(Q_{m,tg}^k\right)_{\text{dp}}}} = k_e(1 - \text{RH}_m)\frac{\left(Q_{m,tg}^1\right)_{\text{dp}}+\left(Q_{m,tg}^2\right)_{\text{dp}}+\cdots+\left(Q_{m,tg}^n\right)_{\text{dp}}}{\sqrt{\sum_{k=1}^{n}\left(Q_{m,tg}^k\right)_{\text{dp}}}} \tag{R5}$$

Because Eq. (R3) reflects that there is positive correlation between the evaporation rate and precipitation flux, we assume the individual evaporation rate of tagged convection precipitation from source region $k$ is expressed as:

$$\left(\frac{\partial q_{v,tg}^k}{\partial t}\right)_{\text{dp\_evap}} = \begin{cases} k_e(1 - \text{RH}_m)\frac{\left(Q_{m,tg}^k\right)_{\text{dp}}}{\sqrt{\sum_{k=1}^{n}\left(Q_{m,tg}^k\right)_{\text{dp}}}}, & \text{if } \sum_{k=1}^{n}\left(Q_{m,tg}^k\right)_{\text{dp}} \neq 0, \\ 0, & \text{if } \sum_{k=1}^{n}\left(Q_{m,tg}^k\right)_{\text{dp}} = 0, \end{cases} \tag{R6}$$

In addition, the evaporation rate of convection precipitation is very small compared to the tendency of water vapour in convection process, as reported by Neale et al. (2012). Equation (R6) is sufficient to partition the evaporation rate of each tagged convection precipitation.

Changes in manuscript: The corresponding descriptions have been added in Sect. 2.1 for deep convection and in Sect. 2.2 for shallow convection.

**5. Some of the formulation of tagged water substance in the macrophysics are confusing, especially for the cloud fraction. How can the stratus cloud fraction be composed proportionally of each tagged condensates without mixing?**

Reply: The separate liquid stratus fraction $a_{l,\text{st}}$ is a unique function of grid-mean relative humidity over water, $\bar{u}_l \equiv \bar{q}_v/\bar{q}_{s,w}$, where $\bar{q}_v$ is the grid-mean water vapour specific humidity and $\bar{q}_{s,w}$ is the grid-mean saturation specific humidity over water, which is shown in Eq. (3) of Park et al. (2014):

$$a_{l,\text{st}} = \begin{cases} 1, & \text{if } \bar{u}_l \geq \hat{u}_l, \\ 1-\left[\frac{3}{\sqrt{2}}\left(\frac{\hat{u}_l-\bar{u}_l}{\hat{u}_l-u_{\text{cl}}}\right)\right]^{\frac{2}{3}}, & \text{if } \frac{1}{6}(5\hat{u}_l+u_{\text{cl}}) \leq \bar{u}_l \leq \hat{u}_l, \\ 4\cos\left[\frac{1}{3}\left\{\arccos\left[\frac{3}{2\sqrt{2}}\left(\frac{\hat{u}_l-\bar{u}_l}{\hat{u}_l-u_{\text{cl}}}\right)\right]-2\pi\right\}\right]^2, & \text{if } u_{\text{cl}} \leq \bar{u}_l \leq \frac{1}{6}(5\hat{u}_l+u_{\text{cl}}), \\ 0, & \text{if } \bar{u}_l \leq u_{\text{cl}}, \end{cases} \tag{R7}$$

where in-cloud RH $\hat{u}_l = 1$. $u_{\text{cl}}$ is the critical RH that liquid stratus starts to form and serves as a tuning parameter in CAM5.1, whose values depended on height and surface properties are presented in Park et al. (2014). Then the single-phase (no separate liquid and ice phases) liquid stratus fraction is

$$A_{l,\text{st}} = (1 - A_{\text{cu}})a_{l,\text{st}} \tag{R8}$$

Here $A_{\text{cu}}$ is the total cumulus fraction.

Equation (R7) is a complicated nonlinear function of $\bar{u}_l$, and it is difficult to separately extract the individual tagged liquid stratus fraction. $A_{l,\text{st}}$ is a monotone increasing function of $\bar{u}_l$: the larger MMR of grid-mean tagged water vapour $\bar{q}_{v,tg}^k$, the more contribution to $A_{l,\text{st}}$ from the source region $k$. All the tagged water substances from the source region are assumed to have the identical physical properties and be well-mixed. Thus, we simply allocate the tagged single-phase liquid stratus fraction $A_{l,\text{st},tg}^k$, which depends on the ratio of $\bar{q}_{v,tg}^k$ and the corresponding sum and is expressed as:

$$A_{l,\text{st},tg}^k = \left(\frac{\bar{q}_{v,tg}^k}{\sum_{k=1}^n \bar{q}_{v,tg}^k}\right) A_{l,\text{st}} \tag{R9}$$

The tagged grid-mean liquid stratus condensate $\bar{q}_{l,a,tg}^k$ is calculated in the same way as the grid-mean liquid stratus condensate $\bar{q}_{l,a}$, but $A_{l,\text{st},tg}^k$ is substituted for $A_{l,\text{st}}$:

$$\bar{q}_{l,a,tg}^k = A_{l,\text{st},tg}^k \times q_{l,\text{st}} \tag{R10}$$

Here, $q_{l,\text{st}}$ is the in-stratus liquid water content (LWC).

Similar to $a_{l,\text{st}}$, the separate ice stratus fraction $a_{i,\text{st}}$ is a function of the grid-mean total ice RH over ice, $\bar{v}_i \equiv (\bar{q}_v + \bar{q}_i)/\bar{q}_{s,i}$, where $\bar{q}_i$ is the MMR of grid-mean ice and $\bar{q}_{s,i}$ is the grid-mean saturation specific humidity over ice, as shown in Eq. (4) of Park et al. (2014):

$$a_{i,\text{st}} = \left(\frac{\bar{v}_i - u_{\text{ci}}}{\hat{u}_i - u_{\text{ci}}}\right)^2 \tag{R11}$$

where the in-cloud RH over ice $\hat{u}_i = 1.1$, and the critical RH that ice stratus begins to form $u_{\text{ci}} = 0.80$ in CAM5.1. Similar to $A_{l,\text{st}}$, the single-phase ice stratus fraction is calculated as

$$A_{i,\text{st}} = (1 - A_{\text{cu}})a_{i,\text{st}} \tag{R12}$$

As in the treatment of $A_{l,\text{st},tg}^k$, the tagged ice stratus fraction $A_{i,\text{st},tg}^k$ is computed based on the ratio of grid-mean total tagged ice specific humidity $(\bar{q}_{v,tg}^k + \bar{q}_{i,tg}^k)$ and the corresponding sum since the nonlinearity of the calculation in $a_{i,\text{st}}$, expressed as

$$A_{i,\text{st},tg}^k = \left[\frac{(\bar{q}_{v,tg}^k + \bar{q}_{i,tg}^k)}{\sum_{k=1}^n (\bar{q}_{v,tg}^k + \bar{q}_{i,tg}^k)}\right] A_{i,\text{st}} \tag{R13}$$

The tagged grid-mean ice stratus condensate $\bar{q}_{i,a,tg}^k$ is calculated in the same way as the grid-mean ice stratus condensate $\bar{q}_{i,a}$:

$$\bar{q}_{i,a,tg}^k = A_{i,\text{st},tg}^k \times q_{i,\text{st}} \tag{R14}$$

Here, $q_{i,\text{st}}$ is the in-stratus IWC. Using the same formula as for the calculation of the grid-mean ambient water vapour specific humidity, the tagged grid-mean ambient water vapour specific humidity $\bar{q}_{v,a,tg}^k$ is computed as follows:

$$\bar{q}_{v,a,tg}^k = \bar{q}_{v,tg}^k + \bar{q}_{l,tg}^k + \bar{q}_{i,tg}^k - \bar{q}_{l,a,tg}^k - \bar{q}_{i,a,tg}^k \tag{R15}$$

Though the tagged cloud fractions were assumed to be composed proportionally of each tagged water vapour specific humidity and the grid-mean total tagged ice specific humidity, the summed tendencies of tagged water substances are very close to the corresponding tendencies of original water substances in most of grid points in cloud processes when the adjustment in Sect. 2.7 is not applied (see Fig. S3). Figure S6 also shows that the summed MMRs of tagged water substances are approximated to the MMRs of original water substances in most of grid points. In addition, our results on the contributions of evaporations from land, the North Atlantic Ocean, extended north Indian Ocean, and extended Northwest Pacific to precipitation over Eurasia are close to the results of Numaguti (1999). In the future, we will find a more exact way to partition the tagged cloud fraction.

Changes in manuscript: The corresponding descriptions of tagged water substance in the macrophysics have been added in Sect. 2.3 in the revised manuscript.

**Minor comments:**

**1. It is "Neale et al." rather than "Neal et al." in the text (line 99).**

Reply: Thank you for your suggestion. We have modified this citation in the revised manuscript.

**2. It is "Gettelman" rather than "Gettleman" in the text (line 189, 190, 196).**

Reply: Thank you for your suggestion. We have modified these citations in the revised manuscript.

**3. It is not clear what Figure S7-S10 exactly show. To where and what the vapour tracers supplied from 25 source regions contribute?**

Reply: Thank you for your suggestion. We have rewritten the captions of these figures (Figs. S10–S13) in the supplement of the revised manuscript. Figures S10 and S11 show the contribution of tagged water vapour tracer from each moisture source region defined in Fig. 1 to water vapour content over Eurasia and its surrounding areas in winter and summer, respectively. Figures S12 and S13 show the contribution of tagged precipitation from each moisture source region defined in Fig. 1 to precipitation over Eurasia and its surrounding areas in winter and summer, respectively.

Changes in manuscript: Please see Lines 48–50, Lines 52–54, Lines 56–57, and Lines 59–60 in the supplement of the revised manuscript.

**Relevant references added in this response:**

Arakawa, A.: The cumulus parameterization problem: Past, present, and future, J. Climate, 17(13), 2493–2525, doi:10.1175/1520-0442(2004)017<2493:RATCPP>2.0.CO;2, 2004.

Dai, A.: Precipitation characteristics in eighteen coupled climate models, J. Climate, 19(18), 4605–4630, doi:10.1175/JCLI3884.1., 2006.

Waliser, D. E., Li, J. L. F., Woods, C. P., Austin, R. T., Bacmeister, J., Chern, J., Genio, A. D., Jiang, J. H., Kuang, Z., Meng, H., Minnis, P., Platnick, S., Rossow, W. B., Stephens, G. L., Sun-Mack, S., Tao, W.-K., Tompkins, A. M., Vane, D. G.,

Walker, C., and Wu, D.: Cloud ice: A climate model challenge with signs and expectations of progress, J. Geophys. Res. Atmos., 114(D8), doi:10.1029/2008JD010015, 2009.

Yang, B., Qian, Y., Lin, G., Leung, L. R., Rasch, P. J., Zhang, G. J., McFarlane, S. A., Zhao, C., Zhang, Y., Wang, H., Wang, M., and Liu, X.: Uncertainty quantification and parameter tuning in the CAM5 Zhang-McFarlane convection scheme and impact of improved convection on the global circulation and climate, J. Geophys. Res. Atmos., 118(2), 395–415, doi:10.1029/2012JD018213, 2013.

[revised manuscript text omitted]

$$D\big(q_{l,tg}^k\big) = \left(\frac{q_{t,u,tg}^k}{\sum_{k=1}^n q_{t,u,tg}^k}\right) \times D(q_l), \ D\big(q_{i,tg}^k\big) = \left(\frac{q_{t,u,tg}^k}{\sum_{k=1}^n q_{t,u,tg}^k}\right) \times D(q_i) \tag{11}$$

This ratio is also applied to the calculations of in-cumulus tagged condensates and the production rates of tagged rain/snow by cumulus expulsion of condensates to the environment. Tagged condensate tendencies for compensating subsidence or upwelling, the tagged condensate tendencies due to detrained cloud water and ice without precipitation contribution, and the updraft/penetrative entrainment mass flux of tagged total water are calculated using the same equations for the original water-related quantities in this scheme. Similar to the calculation of the tendency of water vapour, the tendency of tagged water vapour is computed as the difference between the tendency of tagged total water and the tendencies of tagged condensates in non-precipitating processes within the shallow convection scheme.

The shallow convection scheme relates precipitation evaporation rate $\left(\frac{\partial q_v}{\partial t}\right)_{\text{sh\_evap}}$ to shallow convection precipitation flux

$Q_{\text{sh}}$, similar to the deep convection scheme of CAM5.1. Therefore, we use an assumed expression similar to Eq.

(24) to calculate the tagged precipitation evaporation rate at a level $m$:

$$\left(\frac{\partial q_{v,tg}^k}{\partial t}\right)_{\text{sh\_evap}} = \begin{cases} k_e(1 - \text{RH}_m)\frac{\left(Q_{m,tg}^k\right)_{\text{sh}}}{\sqrt{\Sigma_{k=1}^{n}\left(Q_{m,tg}^k\right)_{\text{sh}}}}, & \text{if } \Sigma_{k=1}^{n}\left(Q_{m,tg}^k\right)_{\text{sh}} \neq 0 \\ 0, & \text{if } \Sigma_{k=1}^{n}\left(Q_{m,tg}^k\right)_{\text{sh}} = 0 \end{cases} \qquad (12)$$

where $\left(Q_{m,tg}^k\right)_{\text{sh}}$ is the tagged precipitation flux at the top interface of level $m$.

**2.3 Cloud Macrophysics**

Park et al. (2014) provided a detailed description of CAM5.1's cloud macrophysics, in which cloud fractions, horizontal and vertical overlapping structures of clouds, and net condensation rates of water vapour into cloud droplets and ice are computed. Because the tendencies of water substances caused by cumulus convection have been calculated in deep and shallow convection schemes, we focus on the treatment of the tagged stratus fraction and net condensation rates of tagged water vapour in stratus clouds in this section.

The separate liquid stratus fraction $Aa_{l,\mathrm{st}}$ is a unique function of grid-mean relative humidity (RH) over water, $\bar{u}_l \equiv \bar{q}_v/\bar{q}_{s,w}$

$\bar{u}_l = \overline{q_v}/\overline{q_{s,w}}$, where $\bar{q}_v$ $\overline{q_v}$ is the grid-mean water vapour specific humidity and $\bar{q}_{s,w}$ $\overline{q_{s,w}}$ is the grid-mean saturation specific humidity over water, which is shown in Eq. (3) of Park et al. (2014). Then the single-phase (no separate liquid and ice phases)

liquid stratus fraction is

$A_{l,\mathrm{st}} = (1 - A_{\mathrm{cu}})a_{l,\mathrm{st}}$           (13)

Here $A_{\mathrm{cu}}$ is the total cumulus fraction.

We allocate the tagged liquid stratus fraction $A_{l,\mathrm{st},tg}^k$, which depends on the ratio of grid-mean tagged water vapour specific humidity $\bar{q}_{v,tg}^k$ $\overline{q_{v,tg}^k}$ and the corresponding sum, expressed as:

$A_{l,\mathrm{st},tg}^k = \left(\dfrac{\bar{q}_{v,tg}^k}{\sum_{k=1}^n \bar{q}_{v,tg}^k}\right) A_{l,\mathrm{st}}$ $\dfrac{\overline{q_{v,tg}^k}}{\sum_{k=1}^n \overline{q_{v,tg}^k}} \times A_{l,\mathrm{st}}$

(14)  (5)

The tagged grid-mean liquid stratus condensate $\bar{q}_{l,\mathrm{a},tg}^k$ is calculated in the same way as the grid-mean liquid stratus condensate $\bar{q}_{l,\mathrm{a}}$, but $A_{l,\mathrm{st},tg}^k$ is substituted for $A_{l,\mathrm{st}}$:

$\bar{q}_{l,\mathrm{a},tg}^k = A_{l,\mathrm{st},tg}^k \times q_{l,\mathrm{st}}$         (15)

Here, $q_{l,\mathrm{st}}$ is the in-stratus liquid water content (LWC). This ratio is also used in the computation of tagged in-stratus liquid water content (LWC) $q_{l,\mathrm{st},tg}^k$ and tagged grid-mean ambient LWC $\overline{q_{l,\mathrm{a},tg}^k}$, thus

$q_{l,\mathrm{st},tg}^k = \dfrac{\overline{q_{v,tg}^k}}{\sum_{k=1}^n \overline{q_{v,tg}^k}} \times q_{l,\mathrm{st}}$         (6)

and

$\overline{q_{l,\mathrm{a},tg}^k} = \dfrac{\overline{q_{v,tg}^k}}{\sum_{k=1}^n \overline{q_{v,tg}^k}} \times \overline{q_{l,\mathrm{a}}}$         (7)

Here,  Similar to $Aa_{l,st}$, the ice stratus fraction $Aa_{i,st}$ is a function of the grid-mean total ice RH over ice, $\bar{v}_i \equiv (\bar{q}_v + \bar{q}_i)/\bar{q}_{s,i}$ , where $\bar{q}_i$  is the grid-mean ice specific humidity and $\bar{q}_{s,i}$  is the grid-mean saturation specific humidity over ice, as shown in Eq. (4) of Park et al. (2014).

Similar to $A_{l,st}$, the single-phase ice stratus fraction is calculated as

$$A_{i,st} = (1 - A_{cu})a_{i,st} \tag{16}$$

As in the treatment of $A^k_{l,st,tg}$, the tagged ice stratus fraction $A^k_{i,st,tg}$ is computed based on the ratio of grid-mean total tagged ice specific humidity $(\bar{q}^k_{v,tg} + \bar{q}^k_{i,tg})$ and the corresponding sum:

$$A^k_{i,st,tg} = \left[\frac{(\bar{q}^k_{v,tg} + \bar{q}^k_{i,tg})}{\sum_{k=1}^{n}(\bar{q}^k_{v,tg} + \bar{q}^k_{i,tg})}\right] A_{i,st} \tag{17}$$

The tagged grid-mean ice stratus condensate $\bar{q}^k_{i,a,tg}$ is calculated in the same way as the grid-mean ice stratus condensate $\bar{q}_{i,a}$:

$$\bar{q}^k_{i,a,tg} = A^k_{i,st,tg} \times q_{i,st} \tag{18}$$

Here, $q_{i,st}$ is the in-stratus ice water content (IWC).

$$\overline{A^k_{i,st,tg}} = \frac{(\overline{q^k_{v,tg}} + \overline{q^k_{i,tg}})}{\sum_{k=1}^{n}(\overline{q^k_{v,tg}} + \overline{q^k_{i,tg}})} \times A_{i,st} \tag{8}$$

$$\overline{q^k_{i,st,tg}} = \frac{(\overline{q^k_{v,tg}} + \overline{q^k_{i,tg}})}{\sum_{k=1}^{n}(\overline{q^k_{v,tg}} + \overline{q^k_{i,tg}})} \times q_{i,st} \tag{9}$$

$$\overline{q^k_{i,a,tg}} = \frac{(\overline{q^k_{v,tg}} + \overline{q^k_{i,tg}})}{\sum_{k=1}^{n}(\overline{q^k_{v,tg}} + \overline{q^k_{i,tg}})} \times \overline{q_{i,a}} \tag{10}$$

 Using the same formula as for the calculation of the grid-mean ambient water vapour specific humidity, the tagged grid-mean ambient water vapour specific humidity

$\bar{q}^k_{v,a,tg}$  is computed as follows:

$$\bar{q}^k_{v,a,tg} = \bar{q}^k_{v,tg} + \bar{q}^k_{l,tg} + \bar{q}^k_{i,tg} - \bar{q}^k_{l,a,tg} - \bar{q}^k_{i,a,tg} \qquad \overline{q^k_{v,a,tg}} = \overline{q^k_{v,tg}} + \overline{q^k_{l,tg}} + \overline{q^k_{i,tg}} - \overline{q^k_{l,a,tg}} - \overline{q^k_{i,a,tg}}$$

$$(19\text{1})$$

In CAM5.1, Park et al. (2014) defined the grid-mean net condensation rate of water vapour into liquid stratus condensate $\bar{Q}_l$

as the time change of $\bar{q}_{l,a}$ minus the external forcing (all processes except stratus macrophysics, including stratus microphysics, moisture turbulence, advection, and convection) of cloud droplets $\bar{F}_l$:

$$\bar{Q}_l = \dot{\bar{q}}_{l,a} - \bar{F}_l = A_{l,\mathrm{st}}\dot{q}_{l,\mathrm{st}} + \alpha q_{l,\mathrm{st}}\dot{A}_{l,\mathrm{st}} - \bar{F}_l \qquad (20)$$

where $\dot{\bar{q}}_{l,a}$, $\dot{q}_{l,\mathrm{st}}$, and $\dot{A}_{l,\mathrm{st}}$ are the time tendency of $\bar{q}_{l,a}$, $q_{l,\mathrm{st}}$, and $A_{l,\mathrm{st}}$ during $\Delta t = 1800$ s, respectively. In CAM5.1, $\alpha = 0.1$

is the ratio of newly formed or dissipated stratus to the preexisting $q_{l,\mathrm{st}}$. Similarly, the tagged grid-mean net condensation rate $\bar{Q}_{l,tg}^k$ is calculated as:

$$\bar{Q}_{l,tg}^k = \dot{\bar{q}}_{l,a,tg}^k - \bar{F}_{l,tg}^k = A_{l,\mathrm{st},tg}^k \dot{q}_{l,\mathrm{st}} + \alpha q_{l,\mathrm{st}}\left(R\dot{A}_{l,\mathrm{st}} + A_{l,\mathrm{st}}\dot{R}\right) - \
[revised manuscript text omitted]